# DeciMamba: Exploring the Length Extrapolation Potential of Mamba

**Assaf Ben-Kish[1], Itamar Zimerman[1], Shady Abu-Hussein[1],**
**Nadav Cohen[1], Amir Globerson[1,2], Lior Wolf[1], Raja Giryes[1]**
[1]Tel Aviv University, [2]Google Research

https://github.com/assafbk/DeciMamba

## Abstract

Long-range sequence processing poses a significant challenge for Transformers due to their quadratic complexity in input length. A promising alternative is Mamba, which demonstrates high performance and achieves Transformer-level capabilities while requiring substantially fewer computational resources. In this paper we explore the length-generalization capabilities of Mamba, which we find to be relatively limited. Through a series of visualizations and analyses we identify that the limitations arise from a restricted effective receptive field, dictated by the sequence length used during training. To address this constraint, we introduce *DeciMamba*, a context-extension method specifically designed for Mamba. This mechanism, built on top of a hidden filtering mechanism embedded within the S6 layer, enables the trained model to extrapolate well even without additional training. Empirical experiments over real-world long-range NLP tasks show that *DeciMamba* can extrapolate to context lengths that are significantly longer than the ones seen during training, while enjoying faster inference.

## 1 Introduction

Lengthy sequences, which can span up to millions of tokens, are common in real-world applications including long books, high-resolution video and audio signals, and genomic data. Consequently, developing Deep Learning (DL) sequence models capable of effectively managing long contexts is a critical objective. Transformers (Vaswani et al., 2017), despite their current dominance in general DL tasks, still face challenges in processing long sequences. Specifically, their quadratic complexity in sequence length makes them computationally demanding, restricting the ability to train them over long sequences and very large datasets.

In recent years, substantial efforts have been made in order to tackle this challenge. The most significant advancements include efficient implementations that increase the model's context length during training (Dao et al., 2022; Liu et al.), and context-extension methods (Chen et al., 2023b; Peng et al., 2023b) designed to effectively expand the context after training. However, recent studies suggest that long-range processing is still an unresolved problem (Li et al., 2024a; Liu et al., 2024a).

One promising approach in this domain is the development of attention-free networks with sub-quadratic complexity, which can be trained more efficiently over long sequence data. In a recent line of works (Gu et al., 2021; Gu et al.; Gupta et al., 2022), the family of state-space layers has been introduced. These layers can be seen as theoretically grounded linear RNNs that can be efficiently computed in parallel via convolutions, thanks to a closed-form formulation of their linear recurrent rule. A recent advancement by Gu & Dao (2024) presented Mamba, which builds on top of an expressive variant of SSMs called Selective State-Space Layers (S6). These layers match or exceed the performance of Transformers in several domains, such as NLP (Lieber et al., 2024; Zuo et al., 2024; Pióro et al., 2024; Wang et al., 2024a), image classification (Zhu et al.; Liu et al., 2024b), audio processing (Shams et al., 2024), genomic data (Schiff et al., 2024), and more.

In this paper, we first explore the length-generalization abilities of Mamba and identify that they are relatively limited. Although Mamba layers are theoretically capable of capturing global interactions at the layer level, we show, through a series of visualizations, analyses, and empirical measures, that

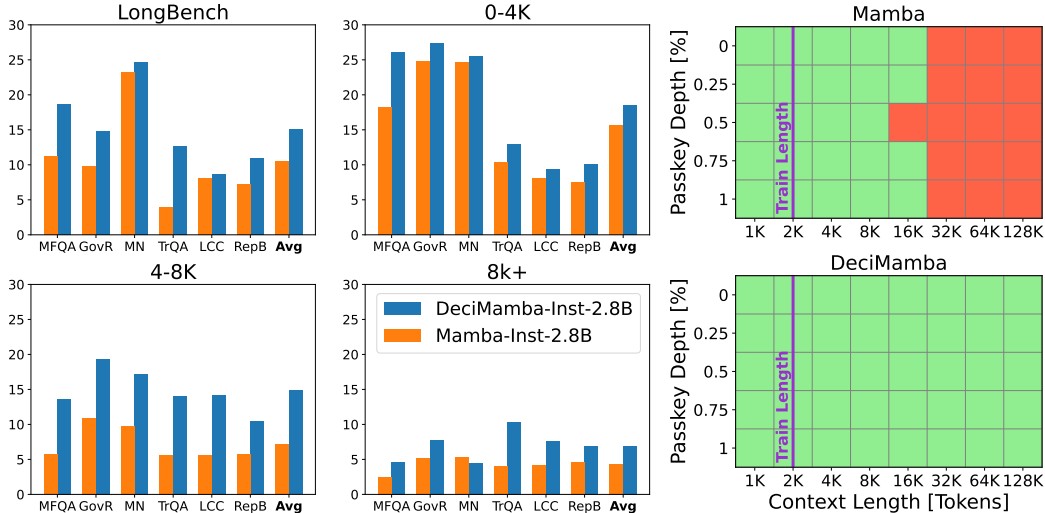

Figure 1: **Improving Mamba Extrapolation with *DeciMamba*.** We present a novel decimation mechanism tailored for Mamba. With our method Mamba is able to process sequences that are much longer than the ones trained on while enjoying reduced inference costs. (**Left**) Zero-shot evaluation of an instruction-tuned 2.8B Mamba model on a subset of LongBench tasks. We show both LongBench and LongBench_e (three length groups: 0-4K, 4-8K, 8k+). MFQA, GovR, MN, TrQA, LCC, RepB, and Avg stand for MultiFieldQA, GovReport, MultiNews, TriviaQA, LCC, RepoBench-p, and Average; (**Right-Top**) Passkey Retrieval for Mamba-130m (**Right-Bottom**) Same for DeciMamba-130m. All models (130m, 2.8b) were trained on lengths of 2K tokens.

the main barrier is Mamba's implicit bias towards sequence lengths that were seen during training, a phenomenon that we call 'limited effective receptive field' (ERF). Next, based on the assumption that long-context data is usually sparse, we present *DeciMamba*, the first context-extension method specifically designed for S6. Our method relies on a dynamic data-dependent pooling method that utilizes a hidden filtering mechanism intrinsic to the Mamba layer. We leverage this mechanism to introduce a global compression operator, which expands Mamba's ERF by discarding unimportant tokens before the S6 layer. In particular, we interpret the norms of the per-token selective recurrent gate ($\Delta_t$) as indicators of each token's importance. This metric allows us to identify the top-k most impactful tokens, enabling direct application of the SSM to these tokens only. The proposed method (Figure 2) significantly increases the effective context length of Mamba by several orders of magnitude while requiring a smaller computational budget.

**Our main contributions** encompass the following three aspects: (i) identify that Mamba has limited length-extrapolation capabilities via thorough experiments in controlled environments. (ii) Through a series of visualizations, analyses, and empirical measures, recognize that although Mamba can theoretically capture global interactions via the recurrent state, its limited ERF prevents significant length-extrapolation; (iii) building on this insight, introduce *DeciMamba*, the first context-extension technique specifically designed for Mamba models. This approach leverages an existing filtering mechanism embedded within the S6 layer. As illustrated in Fig. 1, our method effectively enhances Mamba's length-extrapolation abilities, and is applicable to real-world long-context NLP tasks.

## 2 PRELIMINARIES

Long-range models evolve in two main directions: (i) adapting transformers, the most dominant architecture today, to be more suitable for such tasks; (ii) developing architectures with sub-quadratic complexity in sequence length such as Hyena (Poli et al., 2023), RWKV (Peng et al., 2023a), Hawk (De et al., 2024), xLSTM (Beck et al., 2024), and Mamba, the focus of our paper.

**Mamba.** Given an input sequence $U = (u_1, u_2, \ldots, u_L) \in \mathbb{R}^{L \times d}$ of length $L$ such that $u_i \in \mathbb{R}^d$, a Mamba block with $d$ channels is built on top of the S6 layer via the following formula:

$$G = \sigma(\text{Linear}(U)), \quad X = \text{Conv1D}(\text{Linear}(U)),$$
$$Y = S6(X), \quad O = Y \otimes G \quad (1)$$

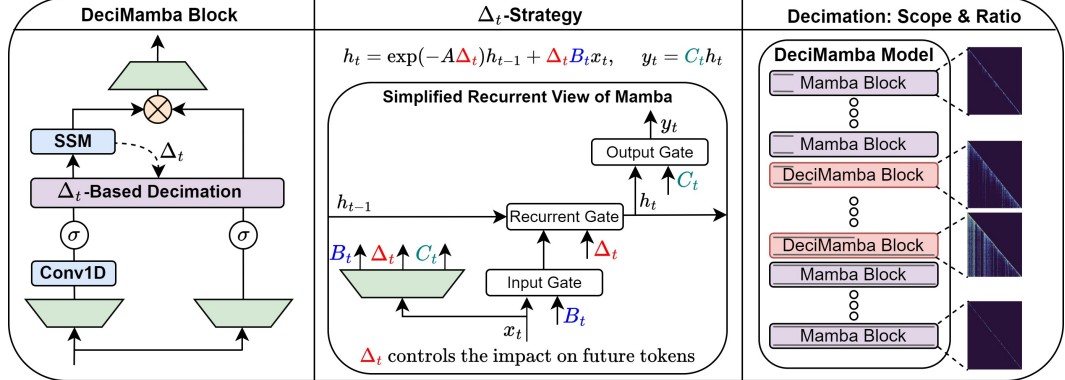

Figure 2: **DeciMamba (Left)** Schematic overview; **(Middle)** By carefully inspecting the recurrent view of Mamba, we revealed the implicit filtering mechanism embedded in the recurrent gate and controlled by $\Delta_t$; **(Right)** Visualization of a *DeciMamba* model. The grey lines represent the sequence length at the input and output of each layer. Layers with a large ERF are decimated.

where $G$ represents the gate branch, $\otimes$ is elementwise multiplication, $\sigma$ is the SILU activation, Linear and Conv1D are standard linear projection and 1-dimensional convolution layers. The S6 layer is based on a time-variant SSM, which can be elaborated by the following recurrent rule:

$$h_t = \bar{A}_t h_{t-1} + \bar{B}_t x_t, \quad y_t = C_t h_t \tag{2}$$

where $X = (x_1, x_2, \ldots, x_L)$ is the input sequence of a representative channel, $\bar{A}_t \in \mathbb{R}^{N \times N}$, $\bar{B}_t \in \mathbb{R}^{N \times 1}$, and $C_t \in \mathbb{R}^{1 \times N}$ are the system, input, and output discrete time-variant matrices, respectively. S6 conditions the discrete time-variant matrices based on the input as follows:

$$\Delta_t = Sft(S_\Delta X_t), B_t = S_B X_t, C_t = (S_C X_t)^T$$
$$\bar{A}_t = \exp(A\Delta_t), \quad \bar{B}_t = B_t \Delta_t \tag{3}$$

such that $\Delta_t$ is the discretization step, *Sft* represents the softplus function, and $S_\Delta, S_B, S_C$ are linear projection layers. Ali et al. (2024) demonstrated that S6 layers, similar to attention models, can be interpreted as data-controlled linear operators. Specifically, the S6 layer computation can be represented using the following linear operator $\alpha$, controlled by the input (via Eq. 3):

$$Y = \alpha X, \quad \alpha_{i,j} = C_i \left( \Pi_{k=j+1}^i \bar{A}_k \right) \bar{B}_j \tag{4}$$

$$\begin{bmatrix} y_1 \\ y_2 \\ \vdots \\ y_L \end{bmatrix} = \begin{bmatrix} C_1 \bar{B}_1 & 0 & \cdots & 0 \\ C_2 \bar{A}_2 \bar{B}_1 & C_2 \bar{B}_2 & \cdots & 0 \\ \vdots & \vdots & \ddots & 0 \\ C_L \Pi_{k=2}^L \bar{A}_k \bar{B}_1 & C_L \Pi_{k=3}^L \bar{A}_k \bar{B}_2 & \cdots & C_L \bar{B}_L \end{bmatrix} \begin{bmatrix} x_1 \\ x_2 \\ \vdots \\ x_L \end{bmatrix} \tag{5}$$

In this formulation, each output $y_i$ is a weighted sum of all inputs, where the 'attention weights' of all inputs $x_j$, i.e., the set $\{\alpha_{i,j}\}_{j=1}^L$, is data-driven. We utilize this perspective to further investigate the effective receptive field of Mamba layers.

**Context Extension & Length Extrapolation.** Several methods were proposed to enhance the effective context length of transformers and improve their extrapolation over longer sequences. Press et al. demonstrated that models built on top of original sinusoidal, rotary (Su et al., 2024), and T5 bias (Raffel et al., 2020) positional encoding have poor length generalization. It proposed to mitigate this issue by incorporating distance-based linear biases into the attention matrix for promoting locality . Kazemnejad et al. (2024) showed that transformers without positional encoding (NoPE) exhibit better length extrapolation capabilities in downstream tasks. Recently, CoPE (Golovneva et al., 2024) utilized context-aware positional encoding and Peng et al. (2023b); Chen et al. (2023a) suggested post-training positional interpolation.

A recent direction involves architectural modifications to pre-trained models followed by short fine-tuning. It includes LongLora (Chen et al., 2023b), which proposes shifted sparse attention, and Landmark Attention (Mohtashami & Jaggi, 2023), which applies attention in chunks and inserts

global unique tokens into the input sequences between those chunks. Our work focuses on taking such an approach for Mamba models, rather than transformers. More related work is in Appendix D. Furthermore, we refer to Appendix D.4, where we motivate the study of Long Context LLMs in parallel to Retrieval Augmented Generation (RAG). Lastly, we note that in order to assess the long range performance of the model, in our experiments we measure perplexity only on the farthest labels of the context window, causing the perplexity values to be larger than their typical values (usually, all tokens in the context window are aggregated).

## 3 EXTRAPOLATION LIMITATIONS OF MAMBA

In this section we explore the length-generalization capabilities of Mamba models. Specifically, we assess the limited effective receptive field (ERF) of S6 layers, a phenomenon that leads to poor information propagation when processing sequences that are longer than the ones trained on. For a detailed description of the ERF, we refer the reader to Appendix D.3. Consequently, we develop methods to measure the portion of the context utilized by the model, and show that it is dictated by the training sequence lengths.

We start by visualizing Mamba's hidden attention (Eq. 4, 5) during extrapolation. In Fig. 3 (left, center) we display the attention matrices of layer 17 of Mamba-130M while *evaluating* sequences of length 2K and 16K (the model trained on the Passkey Retrieval task with sequence lengths of 2K). Notice that while the attention is very dense for the 2K sequence (no extrapolation), the attention for the 16K sequence (extrapolation x8) is much more sparse, and its elements vanish as we move towards the lower rows. This exhibits a limited ERF, as information from the first tokens in the sequence does not propagate to the final tokens in the output sequence.

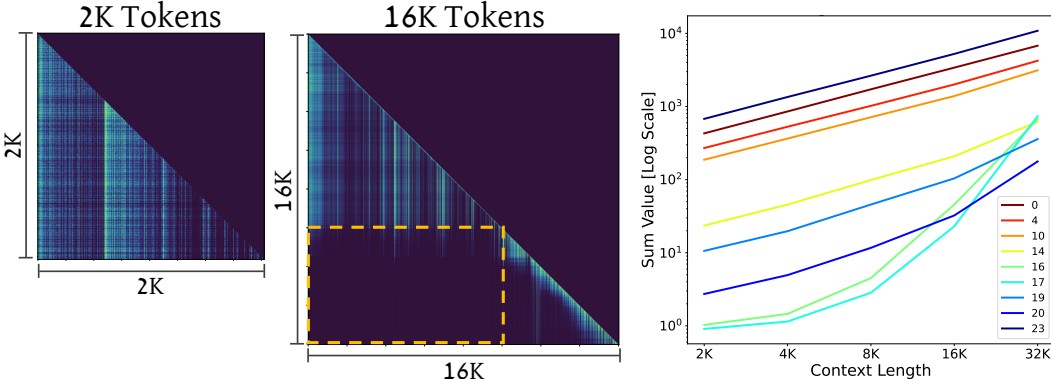

Figure 3: **Detecting and Quantifying Limited ERFs**. (**Center, Left**) Recordings of Mamba Attention Matrices with and without extrapolation (Mamba-130m, layer 17, trained on seq. lengths of 2k). Mamba unintentionally learns a limited ERF during training (highlighted by the dashed rectangle) which disrupts its extrapolation abilities. (**Right**) Quantifying Mamba's Information Loss by Measuring $\sum_{k=2}^{L} \Delta_k$ Divergence. To show Mamba's sensitivity to increasing context lengths we measure the first occasion of information loss, as described in Sec. 3. We observe that in the most semantic layers (16 and 17, see Passkey Retrieval in Sec. 5) $\sum_{k=2}^{L} \Delta_k$ diverges exponentially fast, causing a fast decay in the attention values, leading to limited ERFs like in the center image.

**Measuring ERFs via Mamba Mean Distance.** To quantify how well Mamba utilizes the context during inference, we introduce a quantitative measure called 'Mamba Mean Distance'. This measure is analogous to the receptive field in CNNs and the attention mean distance in transformers described by Dosovitskiy et al. (2020). For a causal transformer model, the attention mean distance for the $i$-th output token is computed by:

$$\mathbb{E}_{j \leq i} \, d(i, j) = \sum_{j \leq i} \tilde{A}_{i,j} \otimes (i - j) \,, \tag{6}$$

where $\tilde{A}$ is the normalized attention matrix that defines a probability distribution over the distances for various tokens. We tailored this measure to S6 by leveraging the implicit attention representation,

which we normalize using the following function:

$$\mathbb{N}(x_1, \cdots, x_L)_j = \frac{|x_j|}{\sum_{k=1}^{L} |x_k|}. \tag{7}$$

As we are usually interested in the last tokens, and as we will see shortly, they are the first to suffer from this phenomenon, we find that it is sufficient to compute our metric for the last token only:

$$\mathbb{E}_{j \leq L} d(L, j) \approx \sum_{j \leq L} \mathbb{N}(\alpha)_{L,j} \otimes (L - j). \tag{8}$$

Fig. 4 visualizes this measure by presenting the 'Mamba Mean Distance' for different hidden attention matrices, depicted by the horizontal distance between the red line and the main diagonal.



Figure 4: **Mamba Mean Distance.** Each panel contains an attention matrix along with its corresponding 'Mamba Mean Distance', depicted by the horizontal distance between the red diagonal line and the main diagonal. The matrices are extracted from a pre-trained model of size 2.8B, trained on the Pile dataset (Gao et al., 2020).

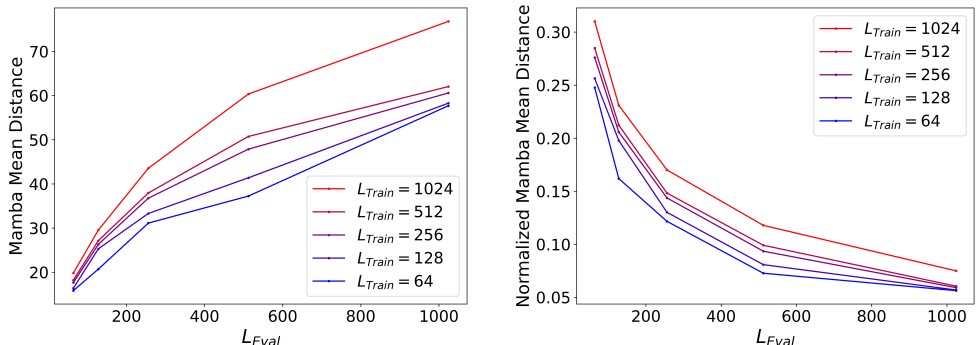

Figure 5: **Mamba Mean Distance Quantifies Effective Context Utilization. (Left)** Mamba Mean Distance as a function of the context length during inference, for various training context lengths. **(Right)** Same, but normalized by the training context length.

By introducing an empirical measure for Mamba's ERF, we can explore it more robustly. Fig. 5 portrays the relationship between the input context length and the Mamba Mean Distance. Here, Mamba-based language models with 80M parameters are trained across various context lengths for next-token prediction on the WikiText-103 benchmark (Merity et al., 2016). We average the Mamba Mean Distance over all channels and layers using 100 test examples. More details on Mamba Mean Distance are in Appendix D.5. Surprisingly, in the left panel of Fig. 5, we see that the Mamba Mean Distance increases with context length. Although it does grow, it is hard to tell whether the growth is proportional to the growth in context length. Therfore, in the right panel, we normalize the Mamba Mean Distance by the sequence length during inference, revealing the exact proportion of context utilized in practice. It is now evident that the context utilization drops dramatically as the evaluated context length increases.

To identify the internal mechanism that leads to this behavior, we return to the analytic expression of Mamba's hidden attention defined in Eq. 5. We observe that the product of transition matrices $\prod_{k=j+1}^{i} \bar{A}_k$ includes more elements as the sequence length $L$ increases, and can be written for the j'th element in the last row as:

$$\prod_{k=j+1}^{L} \bar{A}_k = \prod_{k=j+1}^{L} \exp(A\Delta_k) = \exp(A \sum_{k=j+1}^{L} \Delta_k) \tag{9}$$

We note that this product always converges since $\Delta_t \geq 0$ and $A[i,i] < 0$ by design. Yet, if it converges too fast, the attention element will collapse undesirably. To measure the attention elements' collapse we focus on attention element $\alpha_{L,1}$ due to the following reasons: (i) we are usually interested in the information at the end of the sequence; (ii) this element is the first in the row to collapse, capturing the first occasion of information loss because $\sum_{k=2}^{L} \Delta_k \geq \sum_{k=j+1}^{L} \Delta_k$ for all $L - 1 \geq j \geq 1$, and $A$ is always negative, so the power of its exponent will always be the smallest. As can be seen in Eq. 9, the only input-length dependent value is $\sum_{k=2}^{L} \Delta_k$. Therefore, in Fig. 3 (right), we compute it for varying input lengths L during inference (per layer, averaged over the channels dimension). As can be seen in the figure, the values of $\sum_{k=2}^{L} \Delta_k$ start small and increase exponentially fast in layers 16 and 17, which are the most global layers (Fig. 12). This aligns with the observations in Fig. 3 (left, center) - since these layers require a large ERF, the fast collapse of the attention elements restricts global information propagation, leading to distorted performance. In the other layers, the values of $\sum_{k=2}^{L} \Delta_k$ start large and grow linearly. Since these layers perform local processing (diagonal attention matrices), the fast decay is desired.

To conclude, the limited length-generalization abilities arise due to decay rates $A$ and $\Delta_t$ sums (Eq. 9) that are sufficient for the training data of length $L_{train}$, but not flexible enough to provide effective length-extrapolation for lengths $L_{eval} > L_{train}$. As the transition matrix product converges too fast, we observe a collapse of the attention matrix, specifically in regions that convey information from the beginning of the sequence to its end.

## 4 METHOD: *DeciMamba*

In Sec. 3 we identified that for a pre-trained model, the ERF of Mamba is dictated by the context size used during training $L_{train}$. This creates a blind spot for sequences that exceed $L_{train}$, as dependencies originating in the far input segments are not captured by the model, resulting in poor length generalization. To solve this problem we propose embedding a filtering mechanism within the pre-trained Mamba layers, with no need to re-train the model. The mechanism's core task is to reduce the amount of tokens that the S6 layer processes, and it does so by discarding tokens of fewer importance. The method is described in Fig. 2 and Alg. 1. It encapsulates three aspects: (i) Decimation Strategy, (ii) Decimation Rate, and (iii) Decimation Scope. We refer the reader to Appendix C where we further discuss each hyperparameter's role and selection strategy.

**Decimation Strategy (The Role of $\Delta_t$)**  To pool the most relevant tokens, we must assign each token a relative importance score. We select the $\Delta_t$ parameter as a proxy for this score and motivate our choice by explaining its operation. First, we apply the SSM parametrization to Eq. 2:

$$h_t = \bar{A}_t h_{t-1} + \bar{B}_t x_t = e^{A\Delta_t} h_{t-1} + \Delta_t B_t x_t. \tag{10}$$

Notice that when $\Delta_t \to 0$ the layer discards the input token and preserves the previous hidden state. When $\Delta_t > 0$ the hidden state 'attends' the input token (the attendance is proportional to $\Delta_t$) and adds it to an attenuated version of the previous hidden state (this attenuation is also proportional to $\Delta_t$). We note that this is always the case because $\Delta_t \geq 0$ and $A[i,i] < 0$ by design. Hence, $\Delta_t$ can be interpreted as the controller of the recurrent gate, determining which tokens should impact future tokens. The final importance score for each token is the mean value over all channels of its respective $\Delta_t$ parameter. Note that this selection induces a small interference in Mamba's operation, as Mamba already attempts to ignore input tokens with small $\Delta_t$.

**Decimation Ratio**  For each decimating layer $s$, we propose to keep the Top-$P_s$ tokens with the largest importance scores, where $P_s$ decreases gradually as we go deeper into the network according to the following formula:

$$P_s = L_{base} \cdot \beta^s, \quad \beta \in (0,1), \quad L_{base} \in \mathbb{N}, \tag{11}$$

where $\beta, L_{base}$ are hyper-parameters representing a decay factor that controls the decimation rate (as we progress in depth) and the maximal length of the sequence after the first decimating layer. Sec. E.2 further discusses the selected pooling strategy.

**Decimation Scope (Layer Selection)**  We turn to describe the decimation scope, which defines which layers should use the embedded decimation mechanism. We begin by explaining the guiding principles behind our method, followed by the criteria that reflects those principles. Our goal is to

expand the ERF of a pre-trained Mamba model. Traditionally, DL models focus on various scales of token interactions at different layers. Therefore, a natural design choice is to decimate layers that already focus on long-range dependencies. This approach can potentially increase their ability to learn global dependencies without negatively impacting the layers that are associated with short-term features. As a criterion for measuring the scale embedded within each layer, we measure the ERF of Mamba based on the Mamba Mean Distance defined in Eq. 8. We thus select the layers with the highest Mamba Mean Distance, with the number of selected layers being a hyper-parameter. Note that during inference we perform decimation only in the pre-fill phase. Section E.4 demonstrates that its use in the decoding phase leads to a negligible improvement and thus not required.

**Inference Efficiency** Mamba can be computed in parallel with a prefix-scan or step-by-step like RNNs. For efficient inference, the context is processed in parallel mode (pre-fill), and then the response is generated via auto-regressive decoding in recurrent mode. *DeciMamba* performs pooling only during the pre-fill phase, leveraging the parallel mode to compute the global pooling operation efficiently. During decoding, Mamba and *DeciMamba* have identical forms (no pooling).

**Complexity** In the parallel view of Mamba, the time complexity for processing a sequence of length L, is $O(L \log L)$. In DeciMamba, we first compute the delta-based importance score for each token, identify the top-P elements, and then apply S6 to those P elements. This results in an overall complexity of $O(L + P \log P)$. We ignore the complexity of selecting the top-P elements (which is $O(L \log P)$) since this computation does not depend on the state size N or the number of channels H, making it negligible compared to the other operations in both Mamba and *DeciMamba*.

---

**Algorithm 1** Decimated SSM

---

**Require:** $x : (B - \text{Batch size}, L - \text{Tokens}, D - \text{Channels})$
1: $A : (D, N) \leftarrow Parameter$
2: $B : (B, L, N) \leftarrow S_B(x)$
3: $C : (B, L, N) \leftarrow S_C(x)$
4: $\Delta : (B, L, D) \leftarrow \tau\Delta(Parameter + S_\Delta(x))$
5: $\Delta_P \leftarrow \text{Top-P}(|\Delta|).\text{indices}$
6: $\Delta, A, B, C, x \leftarrow \Delta[\Delta_P], A[\Delta_P], B[\Delta_P], C[\Delta_P], x[\Delta_P]$
7: $A, B : (B, L, D, N) \leftarrow discretize(\Delta, A, B)$
8: return $y : (B, L, D) \leftarrow SSM(A, B, C)(x)$

---

## 5 EXPERIMENTS

We evaluate *DeciMamba* as a context-extension approach across multiple tasks. First, we demonstrate the long-range understanding and retrieval capabilities of our method in the NLP domain. Next, we assess the retrieval capabilities of our method using the Passkey-Retrieval task and examine its language modeling capabilities over PG-19. Finally, we analyze and measure *DeciMamba*'s inference efficiency and provide ablation studies to support our claims.

**LongBench** This benchmark contains a variety of zero-shot long context tasks, such as: Single/Multi Document QA, Summarization, Coding, etc. We evaluate an instruction-tuned Mamba-2.8b (model: 'xiuyul/mamba-2.8b-zephyr' from huggingface) with and without *DeciMamba* (Zero-Shot) and report the results in Table 1. 0-4k, 4-8k, 8k+ are the scores for each context length group in LongBench-E, LB is the LongBench score, and 'N/A' means that the task does not exist in LongBench-E. In almost all tasks, DeciMamba improves the performance of the model for all context lengths. Most noticeable are the improvements at 4-8k, where Mamba starts suffering from ERFs, while DeciMamba extends its performance significantly. E.g. in TriviaQA we improve the LB result from 3.93 to 12.61, a performance gain of 220%, and improve each context length group by 24% (0-4k), 150% (4-8k), and 152% (8k+) respectively. Despite the improvement, when DeciMamba is applied to zero-shot scenarios it does not completely prevent degradation when the context length increases, motivating future research of this behavior. See Sec. A.3 for more details.

**Document Retrieval** In this task the model receives a query and $N_{docs}$ randomly assorted documents, and its objective is to return the id of the matching document. Our data is sampled from SQuAD v2 (Rajpurkar et al., 2018). We train both Mamba-130m and *DeciMamba*-130m to retrieve out of $N_{docs} = 11$ documents (about 2k tokens), and evaluate their performance with

Table 1: **LongBench.** The results are for an instruction-tuned Mamba-2.8b model with and without DeciMamba (Zero-Shot). Avg Len = Average Length in words.

| Type (Metric) | Benchmark | Avg Len | Mamba | | | | DeciMamba | | | |
|---|---|---|---|---|---|---|---|---|---|---|
| | | | 0-4k | 4-8k | 8k+ | LB | 0-4k | 4-8k | 8k+ | LB |
| MultiDoc-QA (F1) | 2wikimqa | 4887 | 7.04 | 1.58 | 0.6 | 3.92 | **11.54** | **6.2** | **3.08** | **9.06** |
| MultiDoc-QA (F1) | Hotpotqa | 9151 | 4.7 | 1.18 | 0.18 | 1.45 | **8.29** | **3.75** | **3.19** | **4.46** |
| MultiDoc-QA (F1) | Musique | 11214 | N/A | N/A | N/A | 0.85 | N/A | N/A | N/A | **1.73** |
| SingleDoc-QA (F1) | Narrative QA | 18409 | N/A | N/A | N/A | 0.87 | N/A | N/A | N/A | **1.74** |
| SingleDoc-QA (F1) | Qasper | 3619 | 6.91 | 4.4 | 1.36 | 5.97 | **8.75** | **9.2** | **2.41** | **8.91** |
| SingleDoc-QA (F1) | Multifield QA | 4559 | 18.26 | 5.71 | 2.45 | 11.16 | **26.05** | **13.67** | **4.63** | **18.58** |
| Summarizing (Rouge-L) | GovReport | 8734 | 24.76 | 10.9 | 5.2 | 9.84 | **27.4** | **19.38** | **7.81** | **14.86** |
| Summarizing (Rouge-L) | QMSum | 10614 | N/A | N/A | N/A | **8.18** | N/A | N/A | N/A | 7.08 |
| Summarizing (Rouge-L) | MultiNews | 2113 | 24.58 | 9.79 | **5.34** | 23.15 | **25.45** | **17.2** | 4.43 | **24.58** |
| Few-Shot (F1) | TriviaQA | 8209 | 10.38 | 5.59 | 4.11 | 3.93 | **12.9** | **14.02** | **10.36** | **12.61** |
| Few-Shot (Rouge-L) | SAMSum | 6258 | **9.58** | **7.07** | **7.76** | **8.56** | 9.17 | 6.76 | 6.88 | 7.34 |
| Few-Shot (Accuracy) | TREC | 5177 | 0.0 | 0.0 | 0.0 | 0.5 | **1.0** | 0.0 | 0.0 | 0.5 |
| Code (Edit Sim) | LCC | 1235 | 8.12 | 5.61 | 4.17 | 8.13 | **9.4** | **14.25** | **7.63** | **8.67** |
| Code (Edit Sim) | RepoBench-p | 4206 | 7.52 | 5.74 | 4.63 | 7.15 | **10.08** | **10.49** | **6.86** | **10.96** |
| Synthetic (Accuracy) | Passage Count | 11141 | 2.0 | 0.0 | 0.0 | 0.0 | **3.0** | 0.0 | 0.0 | **0.5** |
| Synthetic (Accuracy) | Passage Ret. en | 9289 | 0.0 | 0.0 | 0.0 | 0.0 | **9.0** | **1.0** | 0.0 | **1.5** |

$N_{docs} \in [11, 160]$ (2k to 32k tokens). The results are presented in Table 2 (Retrieval). While the performance of Mamba and *DeciMamba* is similar when evaluated on sequence lengths close to those used during training, *DeciMamba* gains a clear advantage as the number of documents exceeds this length, achieving improved length generalization. See Appendix A.2 for more details.

**Multi-Document QA.** Next, we stay in a setting similar to Sec. 5 (Document Retrieval), yet increase the level of difficulty by asking the model to answer the query in free text (instead of retrieving the id of the most relevant document). We train Mamba-130m and *DeciMamba*-130m on the same dataset and present the F1 score between the generated response and the ground truth answers in Tab. 2. As in the previous task, when the amount of documents is close to the amount trained on, the models perform quite similarly. When the document amount increases we can see that *DeciMamba* develops a clear advantage. Note that the performance of both Mamba and *DeciMamba* is relatively modest compared to prior work, due to the use of smaller models.

Table 2: **Multi-Document Retrieval and QA**. The table shows the score of Mamba and DeciMamba (*+Deci*) 130m models as we increase the amount of documents during evaluation. For retrieval we show the accuracy and for free-form QA we show the F1 score between the ground truth annotations and the predictions. Both models were trained using 11 documents.

| # Docs: | | 10 | 20 | 40 | 60 | 80 | 100 | 120 | 140 | 160 |
|---|---|---|---|---|---|---|---|---|---|---|
| Retrieval (Acc) | Mamba | **68.3** | 74.0 | 64.7 | 45.3 | 24.7 | 5.7 | 1.0 | 0.3 | 0.3 |
| | +Deci | 67.7 | **77.3** | **68.7** | **65.3** | **49.7** | **37.0** | **26.3** | **16.7** | **5.3** |
| QA (F1) | Mamba | 25.3 | 22.8 | 23.6 | **25.8** | **19.3** | 7.5 | 4.1 | 1.1 | 0.6 |
| | +Deci | **26.2** | **24.4** | **25.3** | 22.74 | 19.1 | **18.1** | **12.9** | **10.0** | **5.0** |

**Passkey Retrieval.** We fine-tune Mamba-130M and *DeciMamba*-130M to retrieve a random 5-digit code hidden in a random location within a contiguous 2K token sample from Wiki-Text (Merity et al., 2016). To test the models' extrapolation abilities, during inference we increase the sequence lengths exponentially from 1K to 128K and record their performance for a variety of passkey locations within the context. Full implementation details are in Appendix A.1. As can be seen in Fig. 1 (right), *DeciMamba*-130m significantly increases the extrapolation abilities of Mamba-130m from 16K to 128K, when trained on sequence lengths of 2k tokens only. Interestingly, we identify that the $\Delta_t$ values in Layer 16 of Mamba-130m capture the exact location of the passkey, as can be seen in Fig. 10. Using this finding, we display the distortion caused to the embedded sequence due to the ERF in Fig. 11. Tokens at locations $< 10k$ have meaningful $\Delta_t$ values, yet tokens at locations $> 10k$ have noisy $\Delta_t$'s, possibly due to the poor information flow caused by the limited ERF.

**Language Modeling** Following Chen et al. (2023b); Mehta et al. (2022), we evaluate our method on long-range language modeling using the PG-19 dataset in both zero-shot and fine-tuned regimes.

*Zero-Shot.* We test our method on the test set of PG-19 using the larger Mamba models (1.4b, 2.8b) in Fig. 6 (right). We observe that the selectivity scores learned during pre-training are quite effective and extend the context naturally without any training for both models. Furthermore, our method can maintain the low perplexity achieved in the short-context region. In the ablations section we further discuss our proposed decimation mechanism and show its benefit w.r.t other alternatives.

*Fine-Tuning.* We display *DeciMamba*'s performance on PG-19 perplexity in Fig. 6 (left). We train both Mamba-130M and *DeciMamba*-130M with a sequence length of 2K and test their extrapolation abilities. The full training details can be found in Sec. A.4 in the appendix. While Mamba can extrapolate to context lengths that are at most x5 times longer than the training sequences, *DeciMamba* can extrapolate to sequences that are about x20 times longer, without any additional computational resources. Furthermore, we plot a lower bound by calculating the perplexity of a Mamba-130M model that was trained on the same context length it evaluates on (each point on the green curve is a different model). We observe that *DeciMamba* is close to the lower bound and diverges from it quite slowly, while utilizing significantly less computational resources. Nevertheless, for train sequences longer than 30K the 'lower bound' models reach an Out Of Memory (OOM) error (Nvidia RTX-A6000, 48GB of RAM), which is caused by Mamba's O(L) training complexity. This further emphasizes the importance of good extrapolation abilities when scaling to longer sequences.

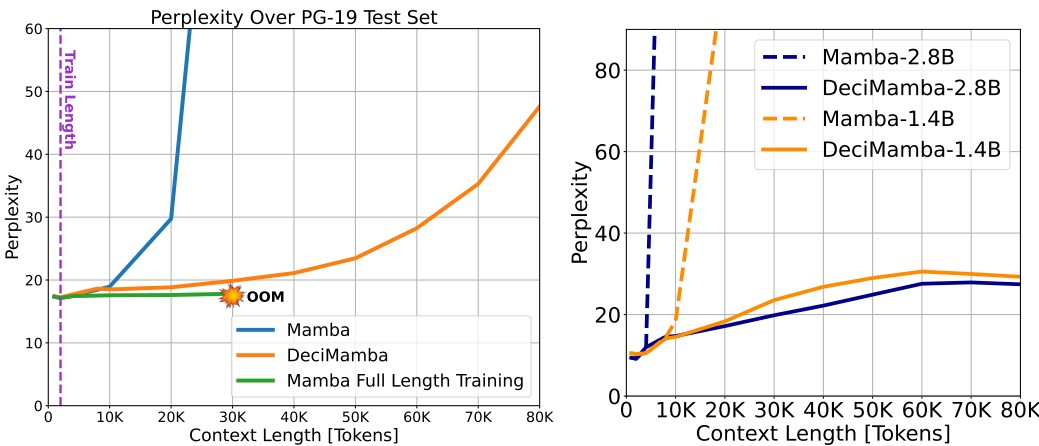

Figure 6: **Perplexity Over PG-19 Test Set**. (**Left**) The dashed purple line shows the train sequence length for Mamba-130m and *DeciMamba*-130m. Each point on the green curve shows a different Mamba-130m model trained on the respective context length ($L_{train} = L_{eval}$); for $L_{train} > 30K$ an Out Of Memory (OOM) error occurs. (**Right**) Zero-Shot perplexity comparison.

**Comparison With Transformers.** We evaluate equivalent Transformer models from the Pythia suite (Biderman et al., 2023) on the experiments described above. We find that vanilla Transformers have inferior length generalization abilities compared to Mamba. Full results are in Appendix B.1.

**Inference Efficiency.** We benchmark both *DeciMamba* and Mamba with a Nvidia RTX A6000 GPU and compare their inference speeds for the Passkey Retrieval task. From Table 3 it is evident that *DeciMamba* is twice faster than the baseline Mamba model, which is expected. This is because the decimation is performed at layer 12 (out of 24 layers in total), hence only half of the blocks process the whole context length L. The other half processes the decimated sequence (of length $P << L$), resulting in a negligible computation time. Table 9 (Appendix) shows that per block, *DeciMamba* does not induce an additional computational overhead when compared to Mamba (first two rows). In addition, we can see that the computation time becomes negligible for blocks that are placed after the first *DeciMamba* block (rows three and four), no matter the input context length.

**Ablation study** The ablation studies can be found in Appendix E. Briefly, these studies examine three key aspects: layer selection, pooling strategy, and decimation mechanism. For layer selection, it seems that the choice of the first decimating layer significantly affects performance, with layer 12

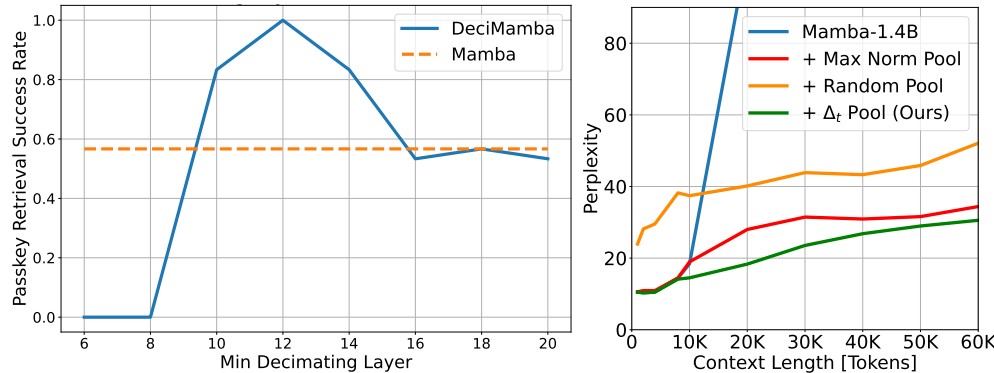

Figure 7: **Ablation Studies**. (**Left**) Model sensitivity to decimation layer selection. We show the score achieved by *DeciMamba*-130M in the Passkey Retrieval task when trained on sequences of length 2K. Each point on the graph is a model's score, while decimating from layer 'Min Decimating Layer' to layer 20. (**Right**) A Zero-Shot perplexity measurement comparing between different pooling mechanisms: ours (based on $\Delta_t$), random and max-norm pooling.

Table 3: **Inference Efficieny.** Inference time (seconds) of the entire model, benchmarked on Nvidia RTX A6000 GPU. *DeciMamba* is about twice as fast when compared to Mamba.

| Model | Context Length | | | | | | |
|---|---|---|---|---|---|---|---|
| | 8K | 16K | 32K | 64K | 128K | 256K | 512K |
| Mamba-130m | 0.17 | 0.3 | 0.69 | 1.27 | 2.12 | 4.03 | 8.17 |
| DeciMamba-130m | **0.14** | **0.19** | **0.31** | **0.59** | **1.08** | **2.01** | **4.22** |

yielding the best results in a Passkey Retrieval task. The pooling strategy study compares a Top-Ps approach with a Top-K% approach, demonstrating that the former is more effective in extending the model's extrapolation abilities. The decimation mechanism study compares various pooling methods, where $\Delta_t$-based pooling performs best, though max-norm pooling also shows promise. In Appendix E.4 we justify our design choice of pooling only during the pre-fill phase of inference. We demonstrate this by ablating an additional pooling mechanism during the decoding phase.

## 6    CONCLUSIONS

In this paper, we explore the length-extrapolation abilities of S6 layers. Our first contribution is the characterization of the ERF of S6. This characterization reveals that the ERF of Mamba is significantly constrained by the context length during training, which is counterintuitive given that Mamba theoretically has an unbounded receptive field due to its recurrent selective memory. Based on these insights, we develop *DeciMamba*, a unique data-dependent compression operator that is built on two key insights: (i) There exists a hidden filtering mechanism within the Mamba layer, manifested by the selective time-steps $\Delta_t$, which can be interpreted as the controller of the recurrent gate. (ii) Long-range and short-range interactions between tokens are captured by different layers in the model, which can be identified using our Mamba Mean Distance metric.

Looking ahead, we plan to explore different transformer context-extension methods, including length-extrapolation PE (Press et al.; Golovneva et al., 2024), hierarchical models, and architectural improvements (Sun et al., 2022). Furthermore, while the techniques described in this paper are designed for Mamba models, a detailed discussion on extending our proposed methods to other models such as Griffin can be found in Appendix F. Finally, we will further use our Mamba Mean Distance Metric to investigate the mechanism which leads to limited ERFs in Mamba.

## 7    LIMITATIONS

Similar to other context-extension methods, our model modifies a pretrained Mamba model but does not propose an improved Mamba architecture to address the underlying issue. Despite empirical evidence demonstrating the effectiveness of our decimation-based method in capturing long-range interactions and its efficiency, the approach can be suboptimal. For example, pooling and compression methods may miss critical information in challenging scenarios. Therefore, designing an improved Mamba variant with enhanced length-generalization abilities that can effectively capture global interactions within a single layer (without pooling the sequence) is an important next step.

ACKNOWLEDGMENTS

This work was supported by a grant from the Tel Aviv University Center for AI and Data Science (TAD), and was partially supported by the KLA foundation. We would like to thank Yael Vinker and Maor Ivgi for fruitful discussions and valuable input which helped improve this work.

## 8 REPRODUCEABILITY STATEMENT

First, we provide the source code used for the key experiments. Second, in appendix A we provide full configurations for all experiments including instructions on how to train and evaluate the models. We also explicitly state all used datasets, models and hardware, and describe the data processing pipeline and exact metrics used in each experiment.

## 9 ETHICS STATEMENT

This work analyzes and improves long-context understanding, which is crucial when deploying LLMs in real-world systems. This improvement is anticipated to have a positive impact on the use of LLMs in society. However, we acknowledge that LLMs can propagate biases. We emphasize the necessity of further research into these biases before our work can be applied reliably beyond the research environment.

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

## A    EXPERIMENTAL DETAILS

All model checkpoints are taken from the Hugging Face Model Hub[1]:

- `state-spaces/mamba-130m`
- `state-spaces/mamba-370m`
- `state-spaces/mamba-790m`
- `state-spaces/mamba-1.4b`
- `state-spaces/mamba-2.8b`
- `iuyul/mamba-2.8b-zephyr`

Our code is based on the official Mamba implementation.[2]

### A.1    PASSKEY RETRIEVAL

Each model is trained for 5 epochs with a learning rate of 1e-4, gradient clipping of 1, batch size of 32 (used batch accumulation) and AdamW optimizer (Kingma & Ba, 2017) with weight decay of 0.1. In each epoch the models train over 6144 sequences of length 2K. For *DeciMamba*-130M we use $L\_base = 2K$, $\beta = 0.5$, $decimating\_layers = [13, \ldots, 21]$, $min\_seq\_len = 20$. Our code is built over an existing version of BABILong (Kuratov et al., 2024).

### A.2    MULTI-DOCUMENT RETRIEVAL AND QA

We train each model with data from SQuAD v2 (Rajpurkar et al., 2018), which provides examples in the form of (Query, Document, Answer). Our training samples have the following form: $N_{docs} \times <Document>$; $<Answer>$, where $<Document>$ can be either the golden document (which holds the answer to the query) or one of $N_{docs} - 1$ randomly sampled documents. $<Answer>$ holds the id of the golden document. In our setting $N_{docs} = 11$, the order of the documents is random, and the query and respective document id are appended to the beginning of each document. During Evaluation we use the same setting but vary the value of $N_{docs}$, between 11 and 160. We note that the length of an average document in SQuAD is a bit smaller than 200 tokens, so our average training sample has about 2,200 tokens, and the evaluation samples vary between 2,200 tokens to 32,000 tokens. We train for two epochs (1500 steps in each), use a learning rate of 1e-4, gradient clipping of 1, batch size of 64 (used batch accumulation), and AdamW optimizer with weight decay of 0.1. We found that the optimal decimation parameters are decimation_layer = 12, $L_{base}$=2000 during training and $L_{base}$=4000 during evaluation. We intentionally decreased Lbase during training so the model could experience decimation during the training period ($L_{train}$ was a bit higher than $L_{base}$), because otherwise the training of DeciMamba and Mamba would have been identical. The displayed results are the average performance over 3 different training seeds.

### A.3    LONGBENCH

We use the 'iuyul/mamba-2.8b-zephyr' instruction-tuned model from huggingface. decimation_layer = 28. $L_{base}$ varies between different benchmarks, between 2000 to 2800.

### A.4    PG-19 PERPLEXITY

We train each model on a total of 100M tokens with a learning rate of 1e-4, gradient clipping of 1, batch size of 250 (used batch accumulation) and AdamW optimizer with weight decay of 0.1. During training we sample a single window from each example and train on it (For the extrapolating models the window length is 2K, for the lower bound models the window length is equal to the context length trained on). During evaluation, for each example we evaluate 10 windows with a maximal constant stride. We evaluate only the last 100 labels in each window, which represent the extrapolation abilities of the model at sequence lengths in the range of $[ctx\_len - 100, ctx\_len]$,

---

[1]https://www.huggingface.co/models
[2]https://github.com/state-spaces/mamba

providing an approximation to the model's performance at the wanted $ctx\_len$. For *DeciMamba-130M* we use $L\_base = 2K$, $\beta = 0.83$, $decimating\_layers = [12, \ldots, 20]$, $min\_seq\_len = 20$. During evaluation we keep the same parameters except setting $L\_base = 8K$. Additionally, in this specific task *DeciMamba* was trained with a similar, yet not identical, Language Modeling (LM) loss. We break the labels sequence (length = 2K) into two chunks. The first 1K labels are trained conditionally on the first 1K tokens of the sequence (like vanilla LM). The last 1K labels are trained conditionally on the whole sequence (2K), and *DeciMamba* was configured to compress the first 1K input tokens. This way we are able to train *DeciMamba* to compress context while training on each label in the sequence, making the training much more efficient. We also experimented with chunking the labels into more than two chunks, but only experienced a slowdown in computation while achieving similar performance. For the lower bound models we had to reduce the amount of training steps in order to constrain the training to 100M tokens. Specifically, for each context length, we followed the following formula: $num\_of\_steps = 100M/(batch\_size * ctx\_len) = 100M/(250 * ctx\_len)$. For the 1.4b model we used Layer 12 for decimation and $L_{base} = 4000$. For the 2.8b model we used Layer 22 for decimation and $L_{base} = 4000$.

# B    ADDITIONAL EVALUATIONS

## B.1    COMPARISON WITH TRANSFORMERS

We find that vanilla Transformers of equivalent size (trained on the same dataset with a similar training recipe), have inferior length generalization abilities compared to Mamba. This is evident in multiple long-context tasks (Tables 4, 5).

Table 4: **Comparison With Transformers - Passkey Retrieval.** The setting is the same as in AppendixA.1. All models were trained on sequences of length 2k. For each context length we test performance for 5 different needle locations, and report the success rate (between 0 and 1). ✓ stands for 100% success, ✗ for 0% success.

| Context Length | 1K | 2K | 4K | 8K | 16K | 32K | 64K | 128K |
|---|---|---|---|---|---|---|---|---|
| Pythia-160M | ✓ | ✓ | ✗ | ✗ | ✗ | ✗ | ✗ | ✗ |
| Mamba-130M | ✓ | ✓ | ✓ | ✓ | 0.8 | ✗ | ✗ | ✗ |
| DeciMamba-130M | ✓ | ✓ | ✓ | ✓ | ✓ | ✓ | ✓ | ✓ |

Table 5: **Comparison With Transformers - Zero-Shot Perplexity.** The setting is the same as in Appendix A.4. All models were pre-trained on sequences of length 2k (we remind that DeciMamba is applied directly without any tuning). inf replaces any perplexity result larger than 100.

| Context Length | 1K | 2K | 4K | 8K | 10K | 20K | 30K | 40K | 50K | 60K | 70K | 80K |
|---|---|---|---|---|---|---|---|---|---|---|---|---|
| Pythia-2.8B | 10.24 | 9.96 | inf | inf | inf | inf | inf | inf | inf | inf | inf | inf |
| Mamba-2.8B | 9.39 | 9.17 | 11.6 | inf | inf | inf | inf | inf | inf | inf | inf | inf |
| DeciMamba-2.8B | 9.39 | 9.17 | 11.98 | 14.58 | 14.73 | 17.17 | 19.83 | 22.2 | 24.89 | 27.57 | 27.89 | 27.43 |
| Pythia-1.4B | 11.64 | 11.32 | inf | inf | inf | inf | inf | inf | inf | inf | inf | inf |
| Mamba-1.4B | 10.51 | 10.31 | 10.5 | 14.43 | 18.4 | inf | inf | inf | inf | inf | inf | inf |
| DeciMamba-1.4B | 10.51 | 10.31 | 10.5 | 14.13 | 14.5 | 18.33 | 23.54 | 26.82 | 28.97 | 30.56 | 29.97 | 29.28 |

For LongBench, the equivalent Transformer model (Pythia-2.8B) repeatedly causes an OOM error on our GPU (A6000, 48GB of RAM).

# C    HYPERPARAMETER SELECTION

In practice, $L_{base}$ is the only parameter we sweep. We typically scan 3 or 4 values that are similar in magnitude to the context length used during training ($L_{train}$). The reason for selecting $L_{base}$ close to $L_{train}$ originates from an assumption on the long-context data: short training sequences and long evaluation sequences have *similar* information content and differ mainly by the *amount of noise* in the sequence. Under this assumption, there are at most $L_{train}$ important tokens; hence, it makes

sense to pool this number of tokens, regardless of the amount of noise. We find this assumption quite reasonable for many long-context tasks, such as retrieval, multi-document question answering, and next-token prediction. For example, in multi-document retrieval/QA, only one document is relevant to the query, regardless of how many random documents we append to the context. Another example is next-token prediction, which is usually very local and does not benefit much from global interactions. Another reason for selecting $L_{base}$ close to $L_{train}$ is that there are no ERF issues when processing sequences of length $L_{train}$, as the global layers (starting from the first decimation layer) are trained on sequences of the same length.

Regarding the number of layers to decimate and the decay rate ($\beta$): The main goal of these parameters is to improve efficiency, not performance. Both parameters introduce the option to compress the sequence further by applying additional DeciMamba layers. We emphasize that this has negligible effects on performance, as shown in the sweeps below for the passkey retrieval task (Tables 6, 7). In addition, this is supported by other results, such as Multi-Document Retrieval and LongBench, which were run with this option disabled.

Table 6: **Decay Rate ($\beta$) Sweep.** BL is the performance of the baseline Mamba model. The setting is the same as in AppendixA.1. We use $L_{base} = 2000$ and 9 decimation layers (13-21).

| $\beta$ | BL | 1 | 0.75 | 0.5 | 0.25 |
|---|---|---|---|---|---|
| Success Rate | 0.55 | 1.0 | 0.975 | 0.975 | 0.975 |

Table 7: **Number of Decimation Layers Sweep.** BL is the performance of the baseline Mamba model. The setting is the same as in Appendix A.1. We use $L_{base} = 2000$ and $\beta = 0.5$ .

| # of Deci Layers | BL | 1 | 2 | 3 | 4 | 5 | 6 | 7 | 8 | 9 |
|---|---|---|---|---|---|---|---|---|---|---|
| Success Rate | 0.55 | 1.0 | 0.975 | 0.925 | 0.925 | 0.975 | 0.925 | 0.95 | 0.95 | 0.975 |

## D    OTHER RELATED WORK

### D.1    LONG RANGE TRANSFORMERS.

Transformers have emerged as highly effective models for various tasks, yet their widespread adoption has been constrained by their limited long-range modeling capabilities. Thus, applying transformers effectively to long-range data remains a central challenge in DL, particularly in NLP. A primary factor in this challenge is that the effective context of transformers is dominated by the context observed during training, which is limited because training LLMs on datasets with billions of tokens across lengthy sequences is computationally demanding. Hence, three main approaches have been developed to tackle this problem: (i) creating efficient variants of transformers that allow an increase in the length of sequences during training. (ii) Context extension methods, which enable training on short sequences and evaluation on long sequences, and finally, (iii) hierarchical models that rely on pooling, chunking, and compression. Despite these extensive efforts, several recent studies indicate that high-quality handling of long text remains an unresolved issue (Liu et al., 2024a; Li et al., 2024a).

### D.2    EFFICIENT TRANSFORMERS.

Over the years, many approaches have been proposed for making transformers more efficient (Tay et al., 2022; Fournier et al., 2023). The two most prominent directions are hardware-aware implementations such as flash-attention (Dao et al., 2022; Dao, 2023) and ring-attention (Liu et al.), which accelerate computations over long sequences by several orders of magnitude. Additionally, developing efficient attention variants with sub-quadratic complexity has become very popular. Two notable examples are Linformer (Wang et al., 2020), which utilizes a low-rank attention matrix, and Performer (Choromanski et al., 2020), a variant that approximates the attention operator through a kernel function.

### D.3 EFFECTIVE RECEPTIVE FIELD (ERF)

The Receptive Field (RF) (LeCun et al., 1998) of a neuron refers to the region of the input space that influences the neuron's output. In traditional CNNs, the RF captures how the area of influence expands as the neuron's depth increases. However, in modern architectures like transformers, which process the entire input context at each layer and can theoretically model interactions between any parts of the input, the concept of RF becomes less informative. In these models, the theoretical receptive field is maximal, rendering the term ambiguous.

To address this, the ERF offers a more practical measure by quantifying the empirical sensitivity of a model to different input regions. The ERF highlights where the model actually focuses its attention and depends not only on the architecture but also on the data and training process. The seminal work on Vision Transformer (ViT) (Dosovitskiy et al., 2020) and Vig & Belinkov (2019) introduced the concept of attention mean distance to measure the ERF, as detailed in Eq. 6. Building upon this, we extend the measurement to Mamba layers in Eq. 8, allowing us to effectively estimate their ERF.

### D.4 WHY USE LONG CONTEXT LLMS INSTEAD OF RETRIEVAL AUGMENTED GENERATION?

Performance-wise, recent studies do not show a conclusive advantage of RAG over Long Context LLMs: Li et al. (2024b), Xu et al. (2024). Interestingly, the common conclusion of all these studies (and others, such as Anthropic's Contextual Retrieval) is that when combined, RAG and Long-Context LLMs have a synergistic effect which improves performance even more. Moreover, each study proposes a different way of combining RAG and Long Context LLMs (directly, switching between RAG and Long Context, augmenting the retriever chunks, etc.) - showing that further research in each area alone, or combined, has potential for an overall improvement in LLM performance.

### D.5 JUSTIFICATION FOR MAMBA MEAN DISTANCE

We are aware of two main approaches for measuring the ERF of Mamba and Attention models: (i) via the attention mean distance or our analogy of Mamba Distance (Eq. 10), and (ii) by measuring the gradient norms over the entire model, which was employed for transformers in Chi et al. (2022) and very recently applied to Mamba models (Jafari et al., 2024) (section 6). The main drawback of the second approach is that it measures the ERF in the context of gradients, which are significantly influenced by the type of loss function, task, and label, complicating the analysis. Additionally, while the first method can be applied to a single layer or a set of layers, the second method becomes less straightforward when applied to a subset of layers, requiring additional considerations and design choices, such as whether to normalize gradients, the use of gradients instead of relevance scores, and the computation of gradients only for attention blocks that learn token interactions or also consider linear and normalization layers. Due to its flexibility and simplicity, as well as being more popular (several examples are Vig & Belinkov (2019); Dosovitskiy et al. (2020); Raghu et al. (2021)), we chose the first method.

## E ABLATIONS

### E.1 LAYER SELECTION

We ablate our layer selection methodology, as it has significant affect on the model's performance (Figure 7, left). Each point on the curve represents the score achieved by a *DeciMamba*-130M model in the Passkey Retrieval task when trained on sequences of length 2K. The only difference between the models is the first decimating layer (x axis). We see that when the decimating layer is too shallow (e.g. layer 8), the model fails completely. As we increase the minimal decimating layer we observe a large increase in performance, until reaching the climax at layer 12. This result aligns with the fact that the $\Delta_t$ distribution is farther from 0 in Mamba's early layers (all tokens are important). We hypothesize that the tokens still don't have a strong representation at this stage, hence it makes less sense to decimate w.r.t their $\Delta_t$ value. After layer 12 the performance of *DeciMamba* starts to drop, and stabilizes at the vicinity of the baseline Mamba model. At this region we start decimating too late, as some long-dependency layers suffer from ERFs when processing the longer Passkey Retrieval sequences. The final layers (21-23) also have larger $\Delta_t$ values, so they should not be

decimated as well. We hypothesize that these layers 'decode' the processed embeddings back into token distributions, so the intermediate representations are yet again not fit for semantic decimation.

### E.2   POOLING STRATEGY

We show the importance of the proposed Top-$P_s$ pooling strategy in the following ablation study. Suppose we select a different pooling strategy, Top-$K\%$, which keeps the top $K\%$ tokens with the largest mean $\Delta_t$ values (over the channels dimension). We train a *DeciMamba*-130m model for the Passkey Retrieval task on sequences of length $L_{train} = 2K$, with $K = 50\%$ and decimation layers $[12, \ldots, 20]$. As can be seen in Fig. 8 (Appendix), Top-$K\%$ extends Mamba's extrapolation abilities from 8K to 32K, yet does not do as well as Top-$P_s$ which extends to 128K. While Top-$K\%$ decimation decreases the sequence length greatly (the output is x512 times shorter), during extrapolation each SSM layer still faces input sequences that are much longer than the ones it has seen during training, as demonstrated in Fig. 9. We further explain this result; During training Layer 17 saw sequences of length 65 ($L_{train} = 2K$, followed by a decimation rate of 50% in each layer from layer 12). During inference it sees sequences of lengths 65 and 1002 (where $L_{eval} = 2K, 32K$, left and right images, respectively). Since the sequence length at layer 17 of the latter (1K) is much longer than the one it saw during training (65), the extrapolation is limited by an ERF (right image, dashed orange shape). This shows the importance of keeping the lengths of the inputs to each decimating SSM layer similar to the ones seen during training, as done by Top-$P_s$.

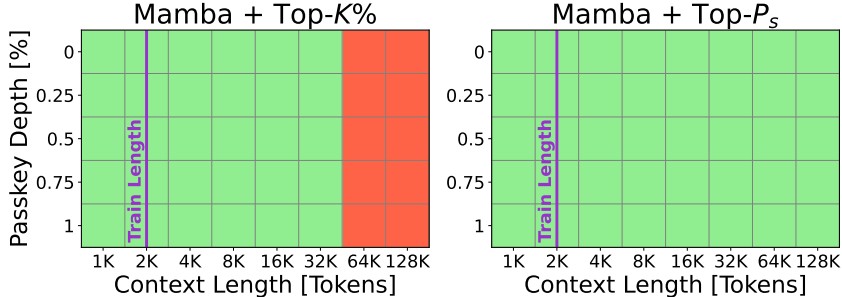

Figure 8: **Pooling Strategy Ablation - Results**. The figure compares two pooling strategies, Top-$P_s$ (ours) and Top-$K\%$. As shown, the Top-$K\%$ approach lags behind the Top-$P_s$ approach, demonstrating that our strategy allows the model to extrapolate to significantly longer sequences. Results are for Mamba-130m.

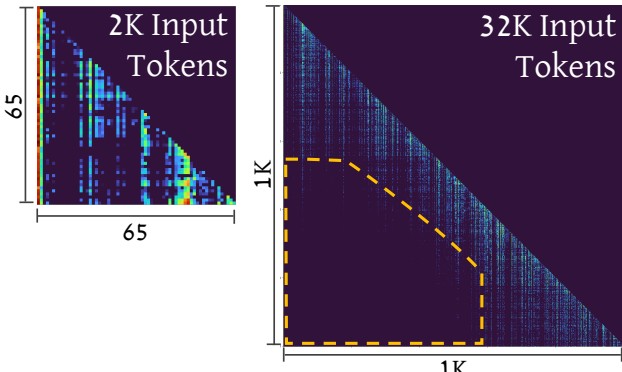

Figure 9: **Pooling Strategy Ablation.** Top-$K\%$ pooling leads to a limited ERF in layer 17. The size of the attention map is affected by the input sequence length: for $L_{eval} = 2K$ layer 17 will process 65 tokens (left) and for $L_{eval} = 32K$ it will process 1K tokens (right). Since $L_{train} = 2K$, layer 17 has only seen training sequences of length 65, therefore suffers from an ERF when $L_{eval}$ increases (right, dashed orange shape).

### E.3 DECIMATION MECHANISM.

To emphasize the benefit of pooling w.r.t the values of $\Delta_t$, we examine two additional decimation mechanisms: max-norm decimation (tokens with maximal norm are kept) and random decimation (tokens are randomly kept). We equip Mamba-1.4b with each of the decimation mechanisms (decimation layer = 12, $L_{base} = 8000$), and compare each model's zero-shot perplexity in Fig. 7 (right). Max-norm pooling achieves a similar trend to $\Delta_t$ pooling (*DeciMamba*), yet performs somewhat worse. The result suggests that $\Delta_t$ is a better candidate for pooling, but also shows that the token's norm also holds some information about its importance. We hypothesize that a method that combines both might achieve better performance than any one alone, but leave this question open for future work. Random pooling induces strong distortion to the embedded sequence, yet achieves better perplexity for longer contexts. This surprising result demonstrates how sensitive Mamba is to limited ERFs - in longer sequences, it is actually better to randomly decimate the sequence rather than process its full length.

### E.4 ADDING ADDITIONAL POOLING DURING DECODING.

By design, DeciMamba performs pooling only during the pre-fill phase of inference. Although the approach is motivated by many previous works (Jiang et al., 2024a; Ge et al., 2024; Jiang et al., 2024b; Li et al., 2023; Jiang et al., 2023; Liu et al., 2024c; Wingate et al., 2022; Mu et al., 2024; Chevalier et al., 2023; Fei et al., 2023; Wang et al., 2024b), we further motivate it by ablating an additional pooling mechanism during the decoding phase. We tested this on the longest generation task in LongBench: GovReport, which involves summarizing long documents with an average length of 10,000 tokens. The additional decimation during decoding was performed by iteratively combining prefill and decoding for each generated chunk, where the chunk size is 50 tokens. Table 8 presents the results and shows that adding decoding pooling has a negligible effect on performance.

A possible explanation for the limited impact of adding decimation during decoding is that, despite being long (about 500 tokens), the generated summaries are relatively short compared to the number of tokens that cause limited ERFs (it needs to be longer than the context length used to train Mamba). Thus, the above suggests that decimation is not needed during decoding. Note also that while in GovReport the generated responses are relatively long, for most of the remaining 15 long-context tasks in LongBench, such as Multi-Document QA, the model is required to generate only 10 to 30 tokens per answer. Therefore, pooling is not required during their decoding phase as well.

Clearly, if a new task emerges where the number of generated tokens is larger than the ERF, it might be useful to add pooling to the decoding phase.

Table 8: **GovReport: Long Summary Generation With and Without Decoding Decimation.** LB is LongBench; 0-4k, 4-8k, 8k+ are the three LongBench-e context length groups; and +DD is DeciMamba with additional Decoding Decimation.

| Model | LB | 0-4k | 4-8k | 8k+ |
|---|---|---|---|---|
| Mamba | 9.8 | 24.8 | 10.9 | 5.2 |
| DeciMamba | **14.9** | **27.4** | **19.4** | 7.8 |
| DeciMamba + DD | 14.7 | 26.6 | 18.4 | **8.14** |

## F EXTENSIONS TO OTHER MODELS

While this paper focuses on exploring the length-generalization abilities and introducing the first context-extension method specifically designed for Mamba models, our proposed techniques are broadly applicable to other architectures. First, the Mamba Mean Distance metric can be generalized to other models that rely on implicit attention. Examples include HGRN (Eq. 5 in Qin et al. (2024)), Griffin (Eq. 14 in De et al. (2024)), RWKV (Eq. 18 in Zimerman et al. (2024)), and others. These tools can provide valuable insights into the ERF of these models, offering a deeper understanding of their length-generalization behavior. Furthermore, the entire context-extension technique introduced in Sec. 4 can be adapted for other models. Specifically, models with per-token data-dependent recurrent gates can leverage these gates to compute token importance scores, enabling a decimation

strategy during pre-fill. For instance, in Griffin, the recurrent gate $r_t$ (Eq. 1 in De et al. (2024) can be used to assign token importance scores, facilitating an effective decimation strategy. Similarly, in HGRN and GLA, the recurrent gate $\lambda_t$ (Eq. 2 in Qin et al. (2024)) and $G_t$ (Eq. 3 in Yang et al. (2023)) accordingly can be interpreted as token importance measures, allowing our approach to be extended to design context-extension techniques for these architectures. By extending these techniques, we provide a foundation for exploring length-generalization in various sequence models.

Table 9: **Inference Time Per Block.** In milliseconds, benchmarked on Nvidia RTX A6000 GPU. *DeciMamba* and Mamba blocks have a similar processing time. Notice how the processing time drops for blocks that are placed after a *DeciMamba* block, no matter the input sequence length.

| Model | Context Length | | | | | | |
|---|---|---|---|---|---|---|---|
| | 8K | 16K | 32K | 64K | 128K | 256K | 512K |
| Mamba Block | **3.2** | **6.1** | **11.6** | **22.7** | 52.5 | 95.9 | 204 |
| DeciMamba Block | 3.3 | **6.1** | 11.8 | 23.1 | **45.9** | **93.1** | **188** |
| Mamba Following DeciMamba | 1.6 | 1.6 | **1.5** | 1.9 | 1.8 | 1.9 | 2.4 |
| DeciMamba Following DeciMamba | **1.3** | **1.4** | **1.5** | **1.5** | **1.5** | **1.5** | **1.5** |

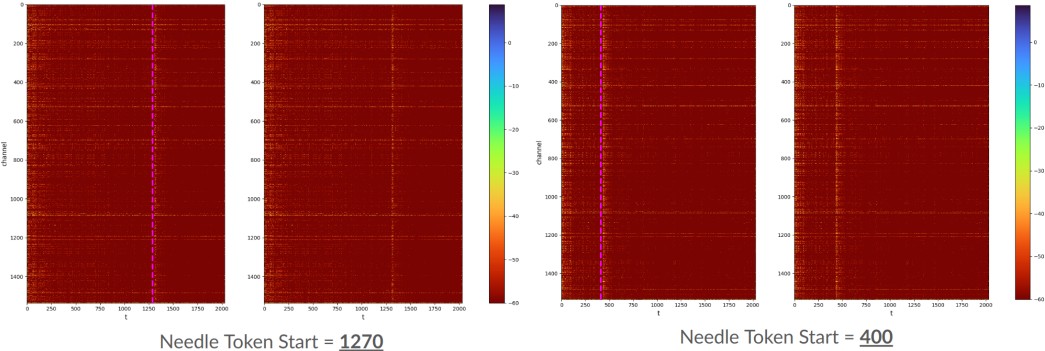

Figure 10: **Demonstrating the 'Importance-Scoring' Abilities of $\Delta_t$.** We evaluate Mamba-130M on the Passkey Retrieval task, and record the values of $\Delta_t$ for all channels of layer 16. Each pair of images is identical, except that the left one marks the location of the passkey with a dashed pink line. The horizontal and vertical axes indicate the token number and the channel respectively. As can be seen from the two cases examined above, the $\Delta_t$ activation captures the needle location successfully, demonstrating the effectiveness of its 'importance scoring' mechanism.

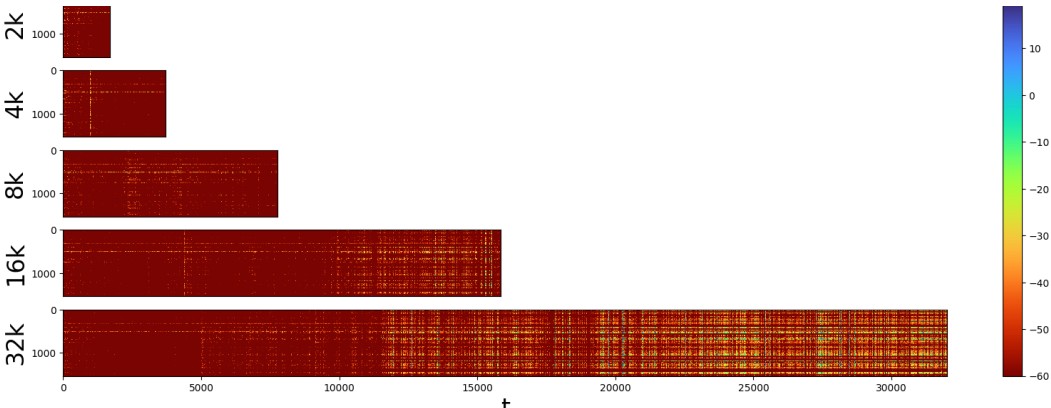

Figure 11: **Measuring the Effects of Limited ERFs.** We show the $\Delta_t$ values across the channels for layer 16 in the Mamba-130M model, examined on different context lengths. The horizontal and vertical axes indicate the token number and the channel respectively. As can be observed from the results above, the passkey can be detected clearly until the ERF ends (for $t > 10K$).

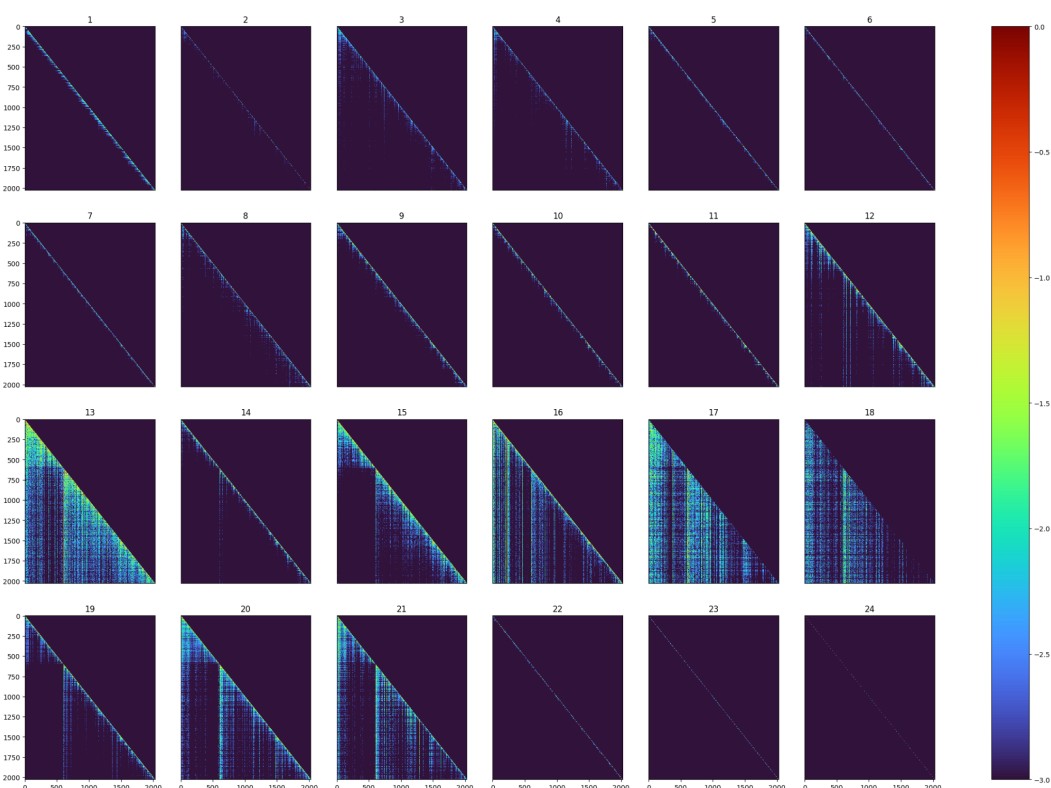

Figure 12: **Normalized Mamba Attention Map**. Displayed in log scale for each layer of the Mamba-130M model.

