# OpenReview forum: "DeciMamba: Exploring the Length Extrapolation Potential of Mamba"
_ICLR.cc/2025/Conference — ICLR 2025 Poster_

### Official Review · Reviewer_txNF · 2024-11-03

**Soundness:** 4
**Presentation:** 4
**Contribution:** 4
**Rating:** 8
**Confidence:** 3

**Summary:**

This paper studies the question of how to understand and address the limitations of length generalization capabilities of the Mamba architecture. The paper introduces ERF, the effective receptive field, and shows that Mamba is biased towards sequence lengths seen during training. The authors then introduce DeciMamba to extend length generalization capabilities by limiting the number of tokens that the S6 layer processes.

**Strengths:**

- The paper introduces the phenomenon of "limited effective receptive field" which could be useful for future work studying the Mamba architecture
- The proposed design is well motivated by the identified ERF phenomenon
- The proposed architecture shows consistent improvement over Mamba in long context range experiments
- The paper includes effective visualizations, especially Figure 2

**Weaknesses:**

- Most of the related work on length generalization is for transformers. Is there any work on this for SSMs? I'm not sure exactly where this falls in the literature.

**Questions:**

- How does the DeciMamba architecture compare to Mamba at short context length tasks? Ie, is there a tradeoff between performance on short and long context tasks?

---

> ### Author Response · Authors · 2024-11-15
> **R4 Rebuttal**
>
> We thank the reviewer for their comments, and provide responses to the raised concerns below.
>
> \
> **Concern 1:**
> ____________________________________________________________________________________________________________
> > Most of the related work on length generalization is for transformers. Is there any work on this for SSMs? I'm not sure exactly where this falls in the literature.
> ____________________________________________________________________________________________________________
> To the best of our knowledge, our work is the first to investigate length generalization in Mamba. We note, however, that several recent follow-up studies [1,2,3] have also explored this topic and were submitted to ICLR 2025.
>
> \
> **Concern 2:**
> ____________________________________________________________________________________________________________
> > How does the DeciMamba architecture compare to Mamba at short context length tasks? Ie, is there a tradeoff between performance on short and long context tasks?
> ____________________________________________________________________________________________________________
> The operation of DeciMamba is identical to that of Mamba in short-context tasks. By definition, DeciMamba pools L_base tokens in the decimation layer. If the sequence length is already shorter than L_base​, it will pool all the tokens, leaving the sequence unchanged.
>
> \
> **References:** \
> [1] LongMamba: Enhancing Mamba's Long-Context Capabilities via Training-Free Receptive Field Enlargement. Anonymous, Submitted to ICLR 2025. \
> [2] MambaExtend: A Training-Free Approach to Improve Long Context Extension of Mamba. Anonymous, Submitted to ICLR 2025. \
> [3] Stuffed Mamba: State Collapse and State Capacity of RNN-Based Long-Context Modeling, Anonymous, Submitted to ICLR 2025.

---

### Official Review · Reviewer_44XJ · 2024-11-03

**Soundness:** 2
**Presentation:** 2
**Contribution:** 2
**Rating:** 3
**Confidence:** 5

**Summary:**

This paper presents the limitations of Mamba in terms of length generalization and proposes an algorithm that combines token dropping with the Mamba architecture to effectively enhance its length generalization capabilities.

**Strengths:**

1. The proposed method is straightforward and can be directly combined with Mamba to improve the length generalization abilities.
2. The authors analyze the shortcomings of Mamba in terms of its length generalization abilities.

**Weaknesses:**

1. The experimental results are limited. The authors only combine the method with the Mamba model and verify the effectiveness of their method. It would be more convincing if the method can be validated on a broader range of SSM-based and linear-attention-based models.
2. The experimental results are not satisfactory. As shown in Table 1, DeciMamba fails to achieve more than a 10% LongBench score on most datasets. In contrast, models with a context window size of only 4k, as reported in the original LongBench paper, often perform better (e.g., Llama-2-7B-chat has an average score of 31 on English tasks). Additionally, for simpler long-context tasks like Passkey, the authors mention that DeciMamba requires fine-tuning to handle them effectively.
3. The presentation of the paper could be improved. For instance, proper citation formats (\citep and \citet) should be used; table captions should appear above the tables; and in Table 1, the DeciMamba entry is missing the LB score for the LCC dataset.

**Questions:**

Please refer to the Weaknesses.

---

> ### Author Response · Authors · 2024-11-15
> **R3 Rebuttal (1/2)**
>
> We appreciate the reviewer’s feedback and provide responses to the raised concerns below.
>
> \
> **Concern 1:**
> ______________________________________________________________________________________________________
> > The experimental results are limited. The authors only combine the method with the Mamba model and verify the effectiveness of their method. It would be more convincing if the method can be validated on a broader range of SSM-based and linear-attention-based models.
> ______________________________________________________________________________________________________
>
> We respectfully disagree with the assertion that the experimental results are limited, as they include multiple model sizes (130M, 1.4B, and 2.8B) across several datasets (PG-19, 16 datasets in LongBench, Passkey-Retrieval, and SQuAD v2) and tasks (language modeling, retrieval, question answering with or without fine-tuning, and summarization), as well as efficiency benchmarking. All of these findings support our claims.
>
> Regarding other models ("...can be validated on a broader range of SSM-based and linear-attention-based models"), we would like to note that Mamba has recently gained significant attention in the field, receiving over 1,000 citations in less than a year, with more than 100 ICLR submissions focused solely on this architecture. We therefore consider our scope to be sufficiently broad and kindly ask the reviewer to reassess this point. Additionally, please note that similar concerns did not arise in the reviews of follow-up studies [4, 5] (also submitted to ICLR '25) that have explored length generalization in Mamba models.
>
> In terms of application to other models, our method is based on the selective, data-dependent delta_t parameter within the selective state-space layers (S6) to measure token importance. Consequently, it cannot be directly applied to models that lack a per-token selective delta_t parameter (though it could be generalized to any data-dependent recurrent gate). Thus, our method cannot be applied to previous SSMs, such as S4, DSS, GSS, S5, Liquid SSMs, and others, which do not incorporate selective mechanisms. Please note that non-selective layers perform significantly worse than Mamba in NLP tasks, making them less relevant for context extension.
>
> Regarding other linear attention models, our method relies on a data-dependent recurrent gate, making it infeasible for non-recurrent models. While some linear attention models could theoretically accommodate our approach (e.g., Griffin, via the r_t parameter in Eq. 1), this is outside the scope of our current work, as we consider Mamba to be broad enough.
>
> To address the reviewer’s concerns, we will add a new technical subsection titled "Extensions to Other Models," detailing how insights from our work can be generalized to other models, such as Griffin, HGRN, RWKV, and others.
>
> \
> **Concern 2:**
> ________________________________________________________________________________________________________________
> > The experimental results are not satisfactory. As shown in Table 1, DeciMamba fails to achieve more than a 10% LongBench score on most datasets. In contrast, models with a context window size of only 4k, as reported in the original LongBench paper, often perform better (e.g., Llama-2-7B-chat has an average score of 31 on English tasks).
> ________________________________________________________________________________________________________________
> Like many other context extension methods [1,2], DeciMamba does not aim to enhance the base model's capabilities (e.g., reasoning). Instead, it primarily helps the model maintain performance on longer sequences.
>
> DeciMamba’s context extension capabilities are clearly demonstrated in the LongBench results. To highlight this, for every benchmark with a score higher than 10% (for either Mamba or DeciMamba), we compute the percentage improvement of DeciMamba:
>
> | Benchmark | 0-4k | 4-8k |
> |---------|---------|-------------|
> | TriviaQA | +24% | +150% |
> | GovReport | +11% | +78% |
> | LCC             | +16% | +154% |
> | MultifieldQA | +43% | +139% |
> | MultinewsQA | +4% | +76% |
> | RepoBench-p | +34% | +82% |
> |**Average**| **+22%** | **+113%** |
>
> In the 0-4k length range (non-long context), DeciMamba improves Mamba by an average of 22%. In sharp contrast, in the 4-8k range (long context), DeciMamba significantly enhances Mamba with an average improvement of 113%.
>
> In the other tasks (e.g. HotpotQA, Qasper, TREC, etc.) Mamba underperforms even in the 0-4k range. Therefore, we do not expect the context extension method to achieve significant performance in these tasks as well.
>
> Finally, Llama-2-7b was trained on 2 trillion tokens and has 7 billion parameters, whereas Mamba-2.8b was trained on 50 billion tokens and has 2.8 billion parameters. Given that in many cases the gap between 2.8b and 7b models holds emerging properties that are crucial for tasks such as LongBench [3], we find this comparison somewhat unfair.

---

> > ### Author Response · Authors · 2024-11-15
> > **R3 Rebuttal (2/2)**
> >
> > **Concern 3:**
> > ________________________________________________________________________________________________________________
> > > ... for simpler long-context tasks like Passkey, the authors mention that DeciMamba requires fine-tuning to handle them effectively.
> > ________________________________________________________________________________________________________________
> >
> > First, we note that it is common to perform fine tuning in the passkey retrieval tasks when evaluating context extension methods [6,7].
> >
> > Second, the released Mamba models are not able to perform passkey retrieval without additional finetuning. **This has nothing to do with our method**, as the baseline model requires fine-tuning as well (for all context lengths, even for 1000 and 2000 tokens).
> >
> > \
> > **Concern 4:**
> > ________________________________________________________________________________________________________________
> > > The presentation of the paper could be improved. proper citation formats (\citep and \citet) should be used; table captions should appear above the tables; and in Table 1, the DeciMamba entry is missing the LB score for the LCC dataset.
> > ________________________________________________________________________________________________________________
> >
> > Thanks for this comment. In the next few days, we will provide a revised manuscript that addresses these issues.
> >
> > Regarding the LCC Dataset results, we apologize for the typo. All of the information is present in the table, but one cell was accidentally shifted to the left (the average length was merged with the benchmark's name). The correct line is as follows (‘M’ for Mamba, ‘D’ For Decimamba):
> >
> > | Benchmark | Avg Len | M. 0-4k | M. 4-8k | M. 8k+ | M. LB | D. 0-4k | D. 4-8k | D. 8k+ | D. LB |
> > |---------|---------|-------------|------|-----|----|------|------|-----|----|
> > | LCC | 1235 | 8.12 | 5.61 | 4.17 | 8.13 | **9.4** | **14.25** | **7.63** | **8.67** |
> >
> >
> >
> > \
> > **References:** \
> > [1] Long Context Compression with Activation Beacon, Zhang et. al 2024 \
> > [2] Extending Context Window of Large Language Models via Positional Interpolation, Chen et. al 2023 \
> > [3] Language Models are Few-Shot Learners, Brown et. al 2022 \
> > [4] LongMamba: Enhancing Mamba's Long-Context Capabilities via Training-Free Receptive Field Enlargement. Anonymous, Submitted to ICLR 2025. \
> > [5] MambaExtend: A Training-Free Approach to Improve Long Context Extension of Mamba. Anonymous, Submitted to ICLR 2025. \
> > [6] Extending Context Window of Large Language Models via Positional Interpolation, S. Chen et. al 2023 \
> > [7] LongLoRA: Efficient Fine-tuning of Long-Context Large Language Models, Y. Chen et. al 2023

---

> > > ### Author Response · Authors · 2024-11-21
> > > **R3 Rebuttal - Remaining Concerns**
> > >
> > > We thank the reviewer again for their feedback. If all concerns are met, we kindly ask the reviewer to increment their score. Nevertheless, if any concerns remain unresolved, we would be happy to further address them.

---

> > > > ### Comment · Reviewer_44XJ · 2024-11-24
> > > >
> > > > 1. My concerns have not been addressed. The authors use a suboptimal Mamba model as the backbone to demonstrate the effectiveness of their method, which is unreasonable. When using a poorly performing backbone, improvement can be easily achieved. Therefore, I request in my review that the authors use more state-of-the-art SSMs and better Mamba models to showcase the effectiveness of their method. However, I do not see new experimental results in the authors' rebuttal.
> > > >
> > > > 2. Additionally, the submitted version contains many basic errors, including:
> > > >    1) Incorrect citation formats.
> > > >    2) Misplaced table captions (even in the latest revision, the positions of the image captions are still incorrect).
> > > >    3) Missing experimental results.
> > > >    These issues indicate that the authors did not thoroughly review their submission.
> > > >
> > > > 3. The proposed method has significant limitations. It requires sorting $\Delta$ and retaining the top k tokens, meaning that DeciMamba must handle all tokens simultaneously and obtain their $\Delta$ information. This undermines the greatest advantage of Mamba, which is its ability to process input sequentially like an RNN. Consequently, DeciMamba loses the characteristic of processing each token/chunk recurrently. Besides, this also implies that DeciMamba's token dropping strategy can only be applied during the pre-filling stage and cannot benefit the decoding stage's token dropping. However, the authors did not discuss and present experimental results for this limitation in the paper.
> > > >
> > > > Therefore, I believe the paper requires more time to include additional experiments and improve the writing. I will maintain my score and strongly defend it.

---

> > > > > ### Author Response · Authors · 2024-11-28
> > > > > **R3 Rebuttal - Second Author Response (1/2)**
> > > > >
> > > > > ## Concern 1:
> > > > > ___
> > > > > > The authors use a suboptimal Mamba model as the backbone to demonstrate the effectiveness of their method, which is unreasonable. When using a poorly performing backbone, improvement can be easily achieved. Therefore, I request in my review that the authors use more state-of-the-art SSMs and better Mamba models
> > > > > ___
> > > > >
> > > > > We thank the reviewer for taking the time to help improve our work. To the best of our knowledge, the only stronger Mamba-based model is Falcon-Mamba-7B, which was trained on context lengths of 8K tokens. Since the LongBench benchmarks involve similar context lengths, Falcon-Mamba-7B does not require length extension for these tasks.
> > > > > Additionally, we note that there are already three follow-up works [12, 13, 14] that test the same models we have used.
> > > > >
> > > > > We emphasize that, in addition to identifying and analyzing the limited Effective Receptive Field (ERF) phenomenon in Mamba, our work also proposes a minimal-intervention solution that significantly improves Mamba's length generalization capabilities across a variety of tasks and model sizes:
> > > > > 1. **Passkey Retrieval:** from 16K token sequences to 128K  token sequences.
> > > > > 2. **Document Retrieval:** from 40 documents to 250 documents (from 8K token sequences to 50K token sequences).
> > > > > 3. **Language Modeling** (e.g., Mamba 2.8B): from 4K token sequences to 60K token sequences.
> > > > > 4. **LongBench:** DeciMamba consistently outperforms the baseline Mamba model across a variety of tasks. In particular, DeciMamba's improvement on LongBench-E in the 4-8K range clearly underscores the impact of our context extension. We respectfully disagree with the reviewer’s assessment that this improvement is either unreasonable or easily achieved.
> > > > >
> > > > > Finally, the existence of the follow up works [12,13,14] further shows that the investigation of token importance scoring in the context of Mamba represents a valuable contribution to the field.
> > > > >
> > > > > .
> > > > > ## Concern 2:
> > > > > ___
> > > > > > Incorrect citation formats, Misplaced table captions, Missing experimental results
> > > > > ___
> > > > >
> > > > > We thank the reviewer for helping us improve our work. We have addressed all the mentioned issues in the revised manuscript.
> > > > > We refer the reviewer to the previous comment, which summarizes all of our experimental results.
> > > > > ## Concern 3:
> > > > > ___
> > > > > > The proposed method has significant limitations. It requires sorting $\Delta_t$ and retaining the top k tokens, meaning that DeciMamba must handle all tokens simultaneously and obtain their $\Delta_t$ information. This undermines the greatest advantage of Mamba, which is its ability to process input sequentially like an RNN. Consequently, DeciMamba loses the characteristic of processing each token/chunk recurrently.
> > > > > ___
> > > > >
> > > > > We refer the reviewer to the official Mamba implementation.
> > > > > During inference, Mamba always processes the context in parallel mode. This is true even for short contexts:
> > > > > [Link](https://github.com/state-spaces/mamba/blob/442fab4b1fd5226c1b5939b37d91ede430b5d1ae/mamba_ssm/modules/mamba_simple.py#L162C9-L205C35)
> > > > >
> > > > > The alternative - processing the whole context in sequential mode - would incur significantly higher inference latencies, as it makes very limited use of the available parallelization hardware.
> > > > > Thus, as shown in the official Mamba code below, only after this phase does Mamba switch to sequential mode to perform auto-regressive prediction.
> > > > > [Link](https://github.com/state-spaces/mamba/blob/442fab4b1fd5226c1b5939b37d91ede430b5d1ae/mamba_ssm/modules/mamba_simple.py#L129C13-L132C27)
> > > > >
> > > > > We highlight that it is not true that DeciMamba limits Mamba’s sequential mode. It is also not true that it limits the processing of each token\chunk recurrently. DeciMamba only operates during the pre-fill phase (parallel mode) and has identical operation to Mamba in the decoding phase (sequential mode). Indeed, we can add our decimation strategy also to the decoding phase but it provides a very limited benefit as we show below and therefore we do not do it.

---

> ### Author Response · Authors · 2024-11-28
> **R3 Rebuttal - Second Author Response (2/2)**
>
> ___
> > DeciMamba's token dropping strategy can only be applied during the pre-filling stage and cannot benefit the decoding stage's token dropping. However, the authors did not discuss and present experimental results for this limitation in the paper.
> ___
>
> We find that focusing exclusively on improving the pre-filling phase does not limit the practical impact of our method. To demonstrate this, we introduce additional pooling in the decoding phase and test this approach on the longest generation task in LongBench, GovReport, which requires summarizing long documents with an average length of about 10,000 tokens.
> We find that adding decoding pooling has a negligible effect on performance.
>
> **GovReport: Long Summary Generation With and Without Decoding Decimation**
> | Model | LB | 0-4k | 4-8k | 8k+ |
> |---------|---------|-------------|------|------|
> | Mamba | 9.8 | 24.8 | 10.9 | 5.2 |
> | DeciMamba |  **14.9** | **27.4** | **19.4** | 7.8 |
> | DeciMamba + DD | 14.7 | 26.6 | 18.4 | **8.14** |
>
> *Where LB is LongBench; 0-4k, 4-8k, 8k+ are the three LongBench-e context length groups; and +DD is DeciMamba with additional Decoding Decimation. The additional decimation during decoding was performed by iteratively combining prefill and decoding for each generated chunk, where the chunk size is 50 tokens.*
>
> A possible explanation for the limited impact of adding decimation during decoding is that, despite being long (about 500 tokens), the generated summaries are relatively short compared to the number of tokens needed to cause limited ERFs. Empirically, for that to happen, the entire sequence should be at least two to three times longer than the length used to train Mamba.
>
> Note also that, while the generated responses in GovReport are relatively long, for most of the remaining long-context tasks in LongBench - such as Multi-Document QA - the model generates only 10–30 tokens per answer. However, if a new task emerges where the generated sequence exceeds the ERF, decimation during decoding can be applied to that task.
>
> In addition, **there is a substantial body of work [1-11] published in top-tier conferences such as ICLR, NeurIPS, and ACL that focuses solely on enhancing the prefill stage** of transformers without modifying the decoding process. Long-context handling during prefilling is a crucial area of research with significant applications, enabling, for instance, domain-specific LLM-based chatbots that can process entire books as context, Multi-Document Question Answering, etc.
>
> We will add this experiment and discussion to the final version of the paper.
>
> \
> In light of the above, we kindly ask the reviewer to reconsider the assessment of this concern as a significant limitation.
>
> \
> **References:** \
> [1] MInference 1.0: Accelerating Pre-filling for Long-Context LLMs via Dynamic Sparse Attention. Jiang et al. NeurIPS 2024 \
> [2] In-context Autoencoder for Context Compression in a Large Language Model. Ge et al. ICLR 2024. \
> [3] LongLLMLingua: Accelerating and Enhancing LLMs in Long Context Scenarios via Prompt Compression. Jiang et al. ACL 2024. \
> [4] Compressing Context to Enhance Inference Efficiency of Large Language Models. Li et al. EMNLP 2023 \
> [5] LLMLingua: Compressing Prompts for Accelerated Inference of Large Language Models. Jiang et al. EMNLP 23 \
> [6] CacheGen: KV Cache Compression and Streaming for Fast Large Language Model Serving. Liu et al. SIGCOMM 2024 \
> [7] Prompt Compression and Contrastive Conditioning for Controllability and Toxicity Reduction in Language Models. Wingate et al. EMNLP 2022 \
> [8] Learning to Compress Prompts with Gist Tokens. Mu et al. NeurIPS 2023 \
> [9] Adapting Language Models to Compress Contexts. Chevalier et al. EMNLP 2023. \
> [10] Extending Context Window of Large Language Models via Semantic Compression. Fei et al. ACL 2024 \
> [11] In-Context Former: Lightning-fast Compressing Context for Large Language Model. Wang et al. EMNLP 2024.
> [12] MambaExtend: A Training-Free Approach to Improve Long Context Extension of Mamba. Anonymous, Submitted to ICLR 2025.
> [13] LongMamba: Enhancing Mamba's Long-Context Capabilities via Training-Free Receptive Field Enlargement. Anonymous, Submitted to ICLR 2025.
> [14] Stuffed Mamba: State Collapse and State Capacity of RNN-Based Long-Context Modeling, Anonymous, Submitted to ICLR 2025

---

> ### Author Response · Authors · 2024-12-02
> **R3 Rebuttal - Remaining Concerns**
>
> We thank the reviewer once again for their valuable feedback, and for engaging with our rebuttal and additional experiments.
>
> We would like to respond to the major concerns you have raised. \
> We emphasize that, in addition to identifying and analyzing the limited Effective Receptive Field (ERF) phenomenon in Mamba, our work also proposes a solution that significantly improves Mamba's length generalization capabilities across a variety of tasks and model sizes (Section 5). We have not seen any reference for these results in your review, despite being 75% of the whole experiments section. \
> Specifically for LongBench, DeciMamba’s context extension capabilities are clearly demonstrated in the previous response, where we show a 113% improvement on context extension tasks. We kindly disagree that this improvement was ‘easily achieved’.
>
> Furthermore, in the previous comment, you expressed concern that DeciMamba’s design limits the sequential mode of Mamba, and that pooling only during pre-fill is a limitation. We explained why the former cannot be true and provided an experiment for the latter, where we show that even for generation-heavy tasks, pre-fill pooling is a robust and effective solution.
>
> We would also like to point out that the new concerns regarding the method’s sequential processing abilities, as well as concerns about the lack of pooling in the decoding phase, were not part of the initial review. Nevertheless, we have made extensive revisions to address all 5 weaknesses that you have identified. The paper has substantially improved in clarity and experimental support as a result of your valuable input:
> 1. Referenced our extensive evaluation suite (Section 5), which tested a large variety of model sizes over 20 benchmarks.
> 2. Fixed citation formats and figure caption locations.
> 3. Showed that limiting pooling to the pre-fill stage is a justified design choice, even in long generation tasks. This is grounded in empirical experiments and by related works from the literature (Appendix D)
> 4. Nevertheless, for our method, pooling can be naturally extended to the decoding phase (Appendix D)
> 5. Highlighted that non-Mamba, ssm-based models are not able to benefit from our method because they do not have a data-dependent, selective $\Delta_t$ parameter.
>
> \
> Should any concerns remain, we would be happy to address them further in the time left. If all concerns have been addressed, we kindly request the reviewer to consider raising their score.

---

### Official Review · Reviewer_gREq · 2024-11-03

**Soundness:** 2
**Presentation:** 2
**Contribution:** 2
**Rating:** 3
**Confidence:** 4

**Summary:**

This paper investigates Mamba's length extrapolation capabilities through a systematic analysis approach. The authors introduce the Mamba Mean Distance metric as a novel way to quantify and analyze Mamba's ability to model long-range dependencies. Their findings reveal inherent limitations in Mamba's capacity to handle longer sequences effectively. To address this challenge, they propose a selective token processing mechanism leveraging the delta t parameter from the Mamba formula. This approach intelligently filters tokens by retaining those with larger delta t values, effectively aligning the input complexity with Mamba's modeling capacity. The method is specifically designed for the prefilling phase and demonstrates promising improvements in Mamba's long-text processing capabilities.

**Strengths:**

- The investigation of token importance scoring in the context of Mamba represents a valuable contribution to the field, especially given the growing interest in alternatives to attention-based mechanisms
- The method achieves substantial improvements while maintaining implementation simplicity, making it readily applicable in practical scenarios

**Weaknesses:**

- Limited scope of application: The method's restriction to the prefilling phase significantly limits its practical impact, especially considering the increasing demand for both long-text prefilling and generation in modern applications
- Potential information loss: The token discarding approach may have unintended consequences in scenarios requiring comprehensive context understanding. This is particularly problematic in tasks like document question-answering, where discarded tokens during prefilling might be crucial for subsequent generation phases
- Incomplete solution to fundamental limitations: While the approach provides a practical workaround, it doesn't address the underlying limitations of Mamba in processing long sequences. A more thorough analysis of the Mamba Mean Distance metric could potentially lead to more fundamental solutions

**Questions:**

- The relationship between distance and performance shown in Fig 1 appears counterintuitive. If Mamba struggles with long-distance relationships, why does the performance degradation manifest primarily in middle lengths rather than showing a clear diagonal boundary from top-left to bottom-right between the red and green regions? Could this suggest a more complex underlying mechanism?
- Given the increasing importance of long-text generation in applications like complex reasoning and creative writing, have the authors explored possibilities of extending this method beyond prefilling to support long-text generation scenarios? It would be benefitial to discuss any challenges the authors foresee in extending the approach to generation tasks.
- The experimental setup shows variations in model selection across different figures and analyses (e.g. Fig 3 and Fig 4). Could the authors provide more detailed justification for these choices and discuss how they might impact the generalizability of the findings?
- Regarding the Mamba Mean Distance results in Fig 4, the findings don't definitively prove that Mamba's long-text capability decreases with length, as this metric wouldn't necessarily scale linearly with sequence length. Are there similar studies for attention-based architectures?

---

> ### Author Response · Authors · 2024-11-15
> **R2 Rebuttal (1/2)**
>
> We appreciate the reviewer’s feedback and provide responses to the raised concerns below.
>
> \
> **Concern 1:**
>  _________________________________________________________________________________________________________
> >Limited scope of application: The method's restriction to the prefilling phase significantly limits its practical impact, especially considering the increasing demand for both long-text prefilling and generation in modern applications
> _________________________________________________________________________________________________________
> We respectfully disagree with the assessment that focusing exclusively on improving the prefilling phase significantly limits the practical impact of our method. **There is a substantial body of work [1-12] published in top-tier conferences such as ICLR, NeurIPS, and ACL that focuses solely on enhancing the prefill stage** of transformers without modifying the decoding process. Long-context handling during prefilling is a crucial area of research with significant applications, enabling, for instance, domain-specific LLM-based chatbots that can process entire books as context.
>
> Moreover, adapting our method to the decoding phase is straightforward. There are two simple extensions:
> 1. **Threshold-Based Pooling Strategy** \
> Instead of using a top-K approach, we can select tokens individually based on whether their \delta_t norms exceed a certain threshold.
> 2. **Combination of Prefill and Decode in a Repetitive Manner for Each Generated Chunk** \
> During the decoding of long sequences, DeciMamba can be applied using the parallel view of Mamba on both the previous history and the newly generated chunk. This allows us to update the current state with the state from DeciMamba. This approach incurs less than twice the computational resources compared to standard decoding, which is negligible [12], but it can enhance long-context capabilities.
>
> A follow-up study [13], which builds on our findings and is also submitted to ICLR '25, employs the threshold-based pooling strategy. Their empirical results suggest that this approach, which extends DeciMamba to decoding, achieves performance similar to that of DeciMamba.
>
> \
> **Concern 2:**
> _________________________________________________________________________________________________________
> > Potential information loss: The token discarding approach may have unintended consequences in scenarios requiring comprehensive context understanding. This is particularly problematic in tasks like document question-answering, where discarded tokens during prefilling might be crucial for subsequent generation phases
> _________________________________________________________________________________________________________
> This isn’t a bug, it’s a feature :)\
> The token discarding approach in our method isn’t about losing information; rather, it’s about selectively focusing on the most relevant context. By design, DeciMamba minimally intervenes in Mamba's operation: it discards tokens with low delta_t values, which the model tries to neglect anyway. The issue is that while the model flags these tokens as unimportant, it does not discard them, so when aggregated, they contribute to an early collapse of the state (Figure 2).
>
> In addition, neural processing naturally emphasizes certain features over others. This is particularly true for state-based models like Mamba, which are designed to compress the context into a fixed-size recurrent state. Given that the number of tokens can vary, modern methods employ data-dependent compression that, by definition, filters out less relevant information in order to prioritize essential content.
>
> Please consider that numerous context-compression techniques for transformers [1-11], published at top-tier venues, demonstrate that this is a robust approach rather than a limitation.
>
> Regarding information loss: while the reviewer raises potential concerns with our method, it is important to clarify that Mamba naturally experiences severe information loss when processing sequences that are longer than those it was trained on, due to its limited ERF (see Section 3). In practice, DeciMamba mitigates this by reintroducing information that may have been forgotten by the model, effectively helping to recover the most important parts of the lost context.
>
> Moreover, as demonstrated in Appendix B.1, DeciMamba maintains performance in tasks with complex inference demands, such as multi-turn dialogue.
>
> Lastly, it is essential to note that DeciMamba is applied only to a subset of layers, excluding those at the beginning of the model, allowing information from unselected tokens to propagate through earlier layers via the remaining tokens.
>
> For all of these reasons, we kindly request that you reconsider this point.

---

> > ### Author Response · Authors · 2024-11-15
> > **R2 Rebuttal (2/2)**
> >
> > **Concern 3:**
> > ________________________________________________________________________________________________________________________
> > > Incomplete solution to fundamental limitations: While the approach provides a practical workaround, it doesn't address the underlying limitations of Mamba in processing long sequences.
> > ________________________________________________________________________________________________________________________
> > We acknowledge that our approach does not fully resolve all ERF-related limitations regarding long-sequence processing and length generalization within the Mamba architecture. However, our work is intended as a first step in this direction. Solving this problem directly is notably challenging and, in some cases, may be theoretically or practically infeasible (e.g. due to the finite size of the state).
> >
> > Additionally, several follow-up studies submitted to ICLR 2025 [13,14] have explored length generalization in Mamba models within a similar regime to ours, and none of them were considered incomplete by reviewers.
> >
> > Finally, as scientific progress is often gradual, we view our contribution, which introduces the following important aspects, as a meaningful advancement in addressing this challenge: (i) shedding light on the limited ERF problem, (ii) supplying an importance-based global pooling solution that has inspired multiple follow-up research efforts [13,14], and (iii) providing a practical method that can be immediately applied to existing Mamba models.
> >
> > For these reasons, we would highly appreciate it if you would consider changing the raised concern from a major weakness to a minor limitation.
> >
> > \
> > **Concern 4:**
> > ________________________________________________________________________________________________________________________
> > > A more thorough analysis of the Mamba Mean Distance metric could potentially lead to more fundamental solutions
> > ________________________________________________________________________________________________________________________
> > Thank you for this suggestion. We will incorporate it into the conclusion of the revised manuscript as an interesting direction for future work.
> >
> > \
> > **References:** \
> > [1] MInference 1.0: Accelerating Pre-filling for Long-Context LLMs via Dynamic Sparse Attention. Jiang et al. NeurIPS 2024 \
> > [2] In-context Autoencoder for Context Compression in a Large Language Model. Ge et al. ICLR 2024. \
> > [3] LongLLMLingua: Accelerating and Enhancing LLMs in Long Context Scenarios via Prompt Compression. Jiang et al. ACL 2024. \
> > [4] Compressing Context to Enhance Inference Efficiency of Large Language Models. Li et al. EMNLP 2023 \
> > [5] LLMLingua: Compressing Prompts for Accelerated Inference of Large Language Models. Jiang et al. EMNLP 23 \
> > [6] CacheGen: KV Cache Compression and Streaming for Fast Large Language Model Serving. Liu et al. SIGCOMM 2024 \
> > [7] Prompt Compression and Contrastive Conditioning for Controllability and Toxicity Reduction in Language Models. Wingate et al. EMNLP 2022 \
> > [8] Learning to Compress Prompts with Gist Tokens. Mu et al. NeurIPS 2023 \
> > [9] Adapting Language Models to Compress Contexts. Chevalier et al. EMNLP 2023. \
> > [10] Extending Context Window of Large Language Models via Semantic Compression. Fei et al. ACL 2024 \
> > [11] In-Context Former: Lightning-fast Compressing Context for Large Language Model. Wang et al. EMNLP 2024. \
> > [12] Just read twice: closing the recall gap for recurrent language models. Arora et al. ICML 2024 (Workshop) \
> > [13] LongMamba: Enhancing Mamba's Long-Context Capabilities via Training-Free Receptive Field Enlargement. Anonymous, Submitted to ICLR 2025. \
> > [14] MambaExtend: A Training-Free Approach to Improve Long Context Extension of Mamba. Anonymous, Submitted to ICLR 2025.

---

> > > ### Author Response · Authors · 2024-11-21
> > > **R2 Rebuttal - Remaining Concerns**
> > >
> > > We thank the reviewer again for their feedback. If all concerns are met, we kindly ask the reviewer to increment their score. Nevertheless, if any concerns remain unresolved, we would be happy to further address them.

---

> > > > ### Comment · Reviewer_gREq · 2024-11-24
> > > >
> > > > Thank you for your detailed response.
> > > >
> > > > - While it is promising to see that the threshold-based pooling strategy has been explored in a follow-up study [13], relying on unpublished or concurrently submitted work to address core limitations in the current submission is not ideal.
> > > > - While the rebuttal provides a compelling theoretical rationale for the token discarding approach, additional empirical evidence is necessary to fully address concerns about its practical implications for tasks requiring comprehensive context understanding, such as long-document QA where documents are followed by questions.
> > > > - Additionally, I encourage the authors to address the questions raised in the "Questions" section alongside the "Weaknesses" section, as doing so would greatly aid in evaluating the submission.

---

> ### Author Response · Authors · 2024-11-28
> **R2 Rebuttal - Second Author Response (1/2)**
>
> ## Concern 1:
> ___
> > While it is promising to see that the threshold-based pooling strategy has been explored in a follow-up study [13], relying on unpublished or concurrently submitted work to address core limitations in the current submission is not ideal.
> ___
> We thank the reviewer for their comment. To address this concern in the best possible way, we introduced additional pooling in the decoding phase and tested this approach on the longest generation task in LongBench, GovReport, which involves summarizing long documents with an average length of 10,000 tokens.
> We find that adding decoding pooling has a negligible effect on performance.
>
> **GovReport: Long Summary Generation With and Without Decoding Decimation**
> | Model | LB | 0-4k | 4-8k | 8k+ |
> |---------|---------|-------------|------|------|
> | Mamba | 9.8 | 24.8 | 10.9 | 5.2 |
> | DeciMamba |  **14.9** | **27.4** | **19.4** | 7.8 |
> | DeciMamba + DD | 14.7 | 26.6 | 18.4 | **8.14** |
>
> *Where LB is LongBench; 0-4k, 4-8k, 8k+ are the three LongBench-e context length groups; and +DD is DeciMamba with additional Decoding Decimation. The additional decimation during decoding was performed by iteratively combining prefill and decoding for each generated chunk, where the chunk size is 50 tokens.*
>
> A possible explanation for the limited impact of adding decimation during decoding is that, despite being long (about 500 tokens), the generated summaries are relatively short compared to the number of tokens needed to cause limited ERFs. Empirically, for that to happen, the entire sequence should be at least two to three times longer than the length used to train Mamba.
>
> Note also that, while the generated responses in GovReport are relatively long, for most of the remaining long-context tasks in LongBench - such as Multi-Document QA - the model generates only 10–30 tokens per answer. However, if a new task emerges where the generated sequence exceeds the ERF, decimation during decoding can be applied to that task.
>
> In light of these results, as well as our previous arguments regarding the applicability of pre-fill compression, we kindly ask the reviewer to reconsider classifying this concern as a core limitation.
>
> .
> ## Concern 2:
> ___
> > While the rebuttal provides a compelling theoretical rationale for the token discarding approach, additional empirical evidence is necessary to fully address concerns about its practical implications for tasks requiring comprehensive context understanding, such as long-document QA where documents are followed by questions.
> ___
>
> We refer the reviewer to Section 5 (LongBench) and, specifically, to Table 1, where we extensively evaluate our method's performance across LongBench’s 16 benchmarks. These benchmarks include tasks such as Long-Document QA, Multi-Document QA, Summarization, Long-Context Few-Shot Learning, and others. The results show that DeciMamba consistently outperforms the baseline Mamba model, demonstrating the effectiveness of our approach. In particular, we highlight DeciMamba’s improvement on LongBench-E in the 4-8K range, which underscores the impact of our context extension.
>
> .
> ## Concern 3:
> ___
> > Addressing the "Questions" subsection
> ___
> We thank the reviewer for this reminder and greatly apologize for missing these questions. We provide full responses below.
>
> ___
> > Regarding the Mamba Mean Distance results in Fig 4, the findings don't definitively prove that Mamba's long-text capability decreases with length, as this metric wouldn't necessarily scale linearly with sequence length. Are there similar studies for attention-based architectures?
> ___
>
> We thank the reviewer for their comment. We agree that the Mamba Mean Distance experiment does not demonstrate that Mamba’s long context capabilities decrease with context length (The performance drop is shown in Section 5: Passkey Retrieval, Multi-Document Retrieval/QA, Perplexity, and LongBench).
>
> The purpose of Figure 4 is to show that the Mamba Mean Distance is an adequate metric for measuring context utilization, as discussed in lines 213-215. We also agree that the figure’s title is misleading, so we have replaced it with: ‘Mamba Mean Distance Quantifies Effective Context Utilization’. Thank you for your constructive feedback, which has improved the paper. We hope this resolves the issue.

---

> > ### Author Response · Authors · 2024-11-28
> > **R2 Rebuttal - Second Author Response (2/2)**
> >
> > ___
> > > The relationship between distance and performance shown in Fig 1 appears counterintuitive. If Mamba struggles with long-distance relationships, why does the performance degradation manifest primarily in middle lengths rather than showing a clear diagonal boundary from top-left to bottom-right between the red and green regions? Could this suggest a more complex underlying mechanism?
> > ___
> >
> >
> > A possible explanation is that the hidden state requires attending to both the passkey and the query. This can be seen in Figure 2 (left), where during normal operation (no extrapolation) the model attends to the passkey in the middle and the query at the beginning. Since the query is always positioned at the start of the sequence, it cannot influence future states beyond a certain point due to the limited ERF. This leads to erroneous hidden states beyond that point.
> >
> > We monitor this behavior in Figure 11 (Appendix). For tokens at positions $t > 10k$, the $\Delta_t$ distribution changes dramatically, regardless of sequence length. Notably, 10k is precisely within the region where performance degrades in the passkey retrieval map (between 8k and 16k).
> >
> > .
> > ___
> > > Given the increasing importance of long-text generation in applications like complex reasoning and creative writing, have the authors explored possibilities of extending this method beyond prefilling to support long-text generation scenarios? It would be beneficial to discuss any challenges the authors foresee in extending the approach to generation tasks.
> > ___
> >
> > In an earlier comment, we mentioned two possible extensions for pooling during decoding (Concern 1 in the first author response):
> > 1. Combination of Prefill and Decode for Each Generated Chunk.
> > 2. Threshold-Based Pooling Strategy.
> >
> > We implemented the first method and tested its impact on long generation tasks, as discussed in the response to Concern 1 in the second author's reply (the introduction of additional pooling in the decoding phase). \
> > Although the generated sequences are long, they still fall within Mamba’s ERF range, so adding decimation to the decoding phase is not necessary.
> > However, if a new task emerges where the generated sequence is significantly longer and exceeds the ERF, decimation during decoding can be applied.
> >
> >
> > .
> > ___
> > > The experimental setup shows variations in model selection across different figures and analyses (e.g. Fig 3 and Fig 4). Could the authors provide more detailed justification for these choices and discuss how they might impact the generalizability of the findings?
> > ___
> >
> > **Justification for Model Size Selection:** \
> > In general, we chose the Mamba-130M model for tasks that require training (such as Multi-Document Retrieval / Passkey Retrieval). In these cases, we selected a model that fits within our GPU (Nvidia RTX-A6000, 48GB of RAM).
> > For other tasks, such as those applicable to Zero-Shot settings (e.g., LongBench), we used the larger Mamba-2.8B model.
> >
> > To clarify, Figure 3 presents the Mamba Mean Distance metric just for illustration. Alternatively, instead of using the Mamba-2.8B model, we could have created the same illustration using hidden attention maps from a Mamba-130M model, as shown in Figure 12 (Appendix).
> >
> > In Figure 4, we trained multiple Mamba models from scratch, each on different context lengths; hence, we used lighter 80M models due to our computational constraints.
> >
> > **Impact on Generalizability:** \
> > We remain cautious when estimating the impact of our method when generalizing to larger models, as LLMs tend to exhibit different emerging properties as their scale increases [1]. However, through our diverse experiment suite, we find that DeciMamba performs well across a wide range of model sizes (130M to 2.8B models). For example, in the language modeling task, we tested a range of model scales: 130M, 1.4B, and 2.8B, and found that DeciMamba can significantly increase the evaluated context in all of them. It even achieved this without any training (Zero-Shot). Another example of DeciMamba's successful operation across different model scales is the use of the Mamba Mean Distance metric, which is used for identifying the relevant decimation layers. Whether applied to the small 130M models or the larger 1.4B/2.8B models, we are consistently able to overcome the limited ERF problem by identifying the global layers correctly.
> > Overall, we attribute the success of our method across different model scales to its simple, minimal-intervention design: DeciMamba only discards tokens with small $\Delta_t$ scores, which the model tries to ignore anyway. Additionally, our method is supported by our analytical analysis (Section 3). It shows that the Mamba architecture is prone to limited ERFs (Equation 9), regardless of the number of model parameters. Discarding unimportant tokens mitigates this issue, as it restores the ERF.
> >
> >
> > \
> > **References:**\
> > [1] Language Models are Few-Shot Learners, Brown et. al 2022

---

> ### Author Response · Authors · 2024-12-02
> **R2 Rebuttal - Remaining Concerns**
>
> We thank the reviewer once again for their valuable feedback, and hope that our response has addressed all of the remaining concerns. \
> We would like to respond to the most major concerns you have raised:
> ___
> > Limited scope of application: The method's restriction to the prefilling phase significantly limits its practical impact
> ___
> > Potential information loss: The token discarding approach may have unintended consequences in scenarios requiring comprehensive context understanding. This is particularly problematic in tasks like document question-answering,
> ___
> In our last response, we showed that neither of these limits our approach in practice, even in long-generation tasks. This was demonstrated through dedicated experiments (Appendix D and Table 1). We kindly request that you reconsider these points, as we have shown evidence that they are not significant weaknesses.
>
> Finally, we have made extensive revisions to address all 5 remaining concerns you raised. As a result of your valuable input, the paper has improved both in clarity and in experimental support:
> 1. We showed that limiting pooling to the pre-fill stage is a justified design choice, even in long generation tasks. This is grounded in empirical experiments and by related works from the literature (Appendix D)
> 2. Nevertheless, for our method, pooling can be naturally extended to the decoding phase (Appendix D)
> 3. We fixed the title of Figure 4 to make it more accurate.
> 4. We provided an explanation for the counter-intuitive relationship between distance and performance in Figure 1 (Passkey Retrieval).
> 5. We justified model size selection in experiments and discussed expected impact on generalizability to very large models.
>
> \
> If all concerns have been addressed, we kindly request the reviewer to consider raising their score. Should any concerns remain, we would be more than happy to address them further in the time remaining.

---

### Official Review · Reviewer_PVqe · 2024-11-03

**Soundness:** 3
**Presentation:** 2
**Contribution:** 3
**Rating:** 6
**Confidence:** 4

**Summary:**

This paper proposes a new method to extend the context length of Mamba. The authors start with an explanation of why Mamba in its original form cannot extend the context length. They propose viewing the S6 block as applying an “attention operation” (Section 2) and based on these attention weights, they compute the Mamba mean distance to measure the effective receptive field of Mamba (Section 3). They basically show that the ERF decreases as the context length at inference time increases and that the main culprit is the fast decrease of the sum of discretization steps. To tackle this issue, they introduce Decimamba, a filtering mechanism that discards tokens of lesser importance (Section 4). They finally show on multiple information retrieval benchmarks that the method better performs on long contexts than Mamba.

**Strengths:**

Overall, I like this paper, I think that Mamba is a very appealing method due to its low inference cost and getting methods that allow extending the context length for Mamba is a very important question. I appreciate the fact that the authors considered diverse benchmarks and the gap between Mamba and Decimamba is pretty consistent in some cases. I also appreciated the scientific approach in the paper that consists in isolating  the problematic component in Mamba and proposing a method to alleviate the issue.

**Weaknesses:**

I have a few concerns regarding this paper that I list below:

- **Hyperparameter choices**: I agree with the fact that the fast decay of the sum of the discrete time steps may explain the lack of length generalization. However, the approach looks a bit hacky in that it introduces multiple novel hyperparameters: the decay factor, the maximal length of the sequence after the first decimating later and the number of layers to decimate. And it does not seem very clear how to make these choices without a gridsearch?
- **Ablations**: have you tried to do an ablation with respect to the number of layers to be decimated or with respect to the decay rate?
Writing should be improved: I found the decimation strategy in Decimamba (Section 4) pretty hard to follow. I think that this should be better explained.
- **Tasks where Decimamba offers much higher gains?**: we see that Decimamba leads sometimes to big improvements (Squad and Passkey retrieval) but on Multi-Document QA, the decay of Mamba seems to also be important. Do you have an explanation/intuition of the type of tasks where Decimamba leads to substantial improvements?



Minor points:

**Improving the introduction**: I think the author should maybe better explain the Decimamba method in the introduction (and maybe add Figure 5 to the introduction?).
- **Lack of length generalization of Mamba**: I was a bit confused about the fact that the authors were surprised by the lack of length generalization of Mamba. I agree with the authors that compared to Transformers, the context cache (the hidden states) do not grow with the context length and thus Mamba can scale the context length to infinity. However, this state is **bounded** and thus the model cannot store all the information it may need . In an information retrieval setting, when the number of documents is large, it is easy to imagine that Mamba has trouble deciding which tokens to store. Anyway, this all to say that the authors should maybe highlight that one of the factors explaining the poor length generalization is the small state space size. I think that this is aligned with the method proposed by the authors in that it looks to decimate some non-important tokens that may be captured in the hidden state.
- **Influence of hidden state size on length generalization**: one thing that i am curious about is: have you tried to study the length generalization of models of similar scale but with one of the models having a bigger state space?
- **Comparison with Transformers**: have you tried to run Transformers in the benchmarks of Section 5? Just curious to see the gap of DeciMamba with Transformers.
- **Out of memory**: Maybe I would add a comment saying that the OOM happens in Figure 6 because the complexity at **training time** is O(L). Most people may have in mind the complexity of Mamba at inference time, that is independent of the context length and may be confused by the OOM you encounter.
- **Typo**: In equation (8), i think the letters "j" and "L" were swapped. It should be $d(L,j)$.

**Questions:**

I asked my questions in the weakness section.

---

> ### Author Response · Authors · 2024-11-21
> **R1 Rebuttal (1/3)**
>
> We thank the reviewer for their detailed comments and suggestions and provide responses to the raised concerns below.
>
> ## Concerns 1+2:
> _________________________________________________________________________________________________________
> > The approach looks a bit hacky in that it introduces multiple novel hyperparameters: the decay factor, the maximal length of the sequence after the first decimating layer and the number of layers to decimate. And it does not seem very clear how to make these choices without a gridsearch.
> _________________________________________________________________________________________________________
> > Have you tried to do an ablation with respect to the number of layers to be decimated or with respect to the decay rate?
> _________________________________________________________________________________________________________
>
> \
> **Maximal Length Parameter (L_base):** \
> In practice, this is the only parameter we sweep. We typically scan 3 or 4 values that are similar in magnitude to the context length used during training (L_train). For example, if the model is trained on 2000 tokens, we check L_base = 1600, 2000, 2400, 2800. We note that the scores do not deviate significantly during the sweep. For instance, across the 16 LongBench benchmarks, the scores deviate by an average of 10%. \
> The reason for selecting L_base close to L_train originates from an assumption on the long-context data: short training sequences and long evaluation sequences have **similar information content** and differ mainly by the **amount of noise** in the sequence. Under this assumption, there are at most L_train important tokens; hence, it makes sense to pool this number of tokens, regardless of the amount of noise. We find this assumption quite reasonable for many long-context tasks, such as retrieval, multi-document question answering, and next-token prediction. For example, in multi-document retrieval/QA, only one document is relevant to the query, regardless of how many random documents we append to the context. Another example is next-token prediction, which is usually very local and does not benefit much from global interactions.
> Another reason for selecting L_base close to L_train is that there are no ERF issues when processing sequences of length L_train, as the global layers (starting from the first decimation layer) are trained on sequences of the same length.
>
> \
> **Number of Layers to Decimate and Decay Rate Parameters:** \
> The main goal of these parameters is to improve efficiency, not performance. In the paper, we introduce the option to compress the sequence further by applying additional DeciMamba layers (by setting number_of_decimation_layers > 1 and decay_factor < 1). We emphasize that this has negligible effects on performance, as shown in the requested ablation below, which sweeps the two parameters for the passkey retrieval task. In addition, this is supported by other results, such as Multi-Document Retrieval and LongBench, which were run with this option disabled. We further emphasized this in the revised manuscript.
>
> \
> **Decay Factor (Beta) Sweep:**
> | Beta: | BL | 1 | 0.75 | 0.5 | 0.25 |
> |---------|---------|-------------|------|------|-----|
> | Success Rate: | 55% | 100% | 97.5% | 97.5% | 97.5% |
>
> BL is the performance of the baseline Mamba model. \
> The setting is the same as the one in the paper. We use L_base=2000 and num_deci_layers = 9.
>
> \
> **Number of Decimation Layers Sweep:**
> | Num Deci Layers: | BL | 1 | 2 | 3 | 4 | 5 | 6 | 7 | 8 | 9 |
> |---------|---------|-------------|------|------|-----|----|------|------|-----|----|
> | Success Rate: | 55% | 100% | 97.5% | 92.5% | 92.5% | 97.5% | 92.5% | 95% | 95% | 97.5% |
>
> BL is the performance of the baseline Mamba model. \
> The setting is the same as the one in the paper. We use L_base=2000 and beta=0.5.
>
>
> .
> ## Concern 3:
> _________________________________________________________________________________________________________
> > Writing should be improved: I found the decimation strategy in Decimamba (Section 4) pretty hard to follow. I think that this should be better explained.
> _________________________________________________________________________________________________________
> We thank the reviewer for their feedback. We will improve the mentioned subsection and clarify the explanation. Specifically, we will clarify at the beginning that our goal is to address ERF issues by generating the most informative hidden state. To achieve this, we focus on prioritizing key content through a data-dependent context compression method and provide a detailed explanation of why Delta_t serves as the most relevant indicator of token importance for future predictions.
> We incorporated this explanation into the revised manuscript (L300-312).
> _________________________________________________________________________________________________________

---

> > ### Author Response · Authors · 2024-11-21
> > **R1 Rebuttal (2/3)**
> >
> > ## Concern 4:
> > _________________________________________________________________________________________________________
> > > we see that Decimamba leads sometimes to big improvements (Squad and Passkey retrieval) but on Multi-Document QA, the decay of Mamba seems to also be important. Do you have an explanation/intuition of the type of tasks where Decimamba leads to substantial improvements?
> > _________________________________________________________________________________________________________
> > We hypothesize that two main factors contribute to the success of DeciMamba:
> >
> > **(1) Tasks in which ERF is the bottleneck:** \
> > Since DeciMamba is a context-extension method, it primarily enhances the model's abilities in tasks where a long context is the main barrier, rather than in other aspects of NLP such as reasoning abilities, domain knowledge, and syntactic and semantic understanding. For example, Mamba is not able to solve the Long ListOps task [1] (Section 2.2.1), even in short context. When applying DeciMamba, there is no change in performance:
> > | L_base | BL | 1K | 1.5K | 2K |
> > |---------|---------|---------|---------|---------|
> > | **Success Rate:** | 19.3% | 19.3% | 19.3% | 19.3% |
> >
> > BL is the performance of the baseline Mamba model. \
> > DeciMamba compresses the context by using different values of L_base, but is unable to improve performance.
> > We follow the setting in [2], where pre-training is applied on the downstream task data followed by fine-tuning on the downstream task. The models train with an average sequence length of 2K tokens.
> >
> >
> > **(2) Tasks with Sparse Data Content** \
> > By design, the pooling operator discards most of the tokens in long sequences. This is effective, for example, in information retrieval tasks, because it helps overcome the limited effective receptive field (ERF) issue while keeping the important information intact. However, this approach may not work as well for tasks that assign similar importance to most of the tokens - e.g., Long ListOps.
> >
> > We will incorporate this discussion into the revised manuscript. \
> > .
> > ## Minor Points
> > **Minor Point 1:**
> > _________________________________________________________________________________________________________
> > > Improving the introduction
> > _________________________________________________________________________________________________________
> > We thank the reviewer for the feedback. We added a more detailed explanation of our method to the introduction. Please refer to the revised manuscript L72-75.
> >
> >
> > \
> > **Minor Point 2+3:**
> > _________________________________________________________________________________________________________
> > > Influence of hidden state size on length generalization
> > _________________________________________________________________________________________________________
> > We refer the reviewer to [3] (Section 3.1 and Figure 2,3), where the relationship between the hidden state size and performance is studied. One interesting finding is that the hidden state's capacity can be reached even with short sequences. This suggests that the state size limits the information density of the context, rather than its length (although in many tasks the two correlate). We emphasize that the limited ERF phenomenon, which is studied in our work, is not related to the state size, as it occurs even in tasks that require minimal memorization, such as Passkey Retrieval.
> > We will incorporate this discussion into the revised manuscript and, specifically, will address the fact that there are additional factors affecting length generalization that are not necessarily related to the limited effective receptive field phenomenon.

---

> ### Author Response · Authors · 2024-11-21
> **R1 Rebuttal (3/3)**
>
> **Minor Point 4:**
> _________________________________________________________________________________________________________
> > Comparison with Transformers
> _________________________________________________________________________________________________________
> We find that vanilla Transformers of equivalent size (trained on the same dataset with a similar training recipe), have inferior length generalization abilities compared to Mamba. This is evident in all long-context tasks tested:
>
>
> \
> **Passkey Retrieval:**
> | Context Length | 1K | 2K | 4K | 8K | 16K | 32K | 64K | 128K |
> |---------|---------|-------------|------|------|-----|----|------|------|
> | Pythia-160M | 100% | 100% | 0% | 0% | 0% | 0% | 0% | 0% |
> | Mamba-130M | 100% | 100% | 100% | 100% | 80% | 0% | 0% | 0% |
> | DeciMamba-130M | 100% | 100% | 100% | 100% | 100% | 100% | 100% | 100% |
>
> The setting is the same as the one in the paper. All models were trained on sequences of length 2k. For each context length we test performance for 5 different needle locations.
>
> \
> **MultiDocument Retrieval:**
> | Num of Docs | 10 | 20 | 40 | 60 | 80 | 100 | 120 | 140 | 160 | 180 | 200 | 250 | 300 |
> |---------|---------|-------------|------|------|-----|----|------|------|------|-----|----|------|------|
> | Pythia-160M | 69% | 0% | 0% | 0% | 0% | 0% | 0% | 0% | 0% | 0% | 0% | 0% | 0% |
> | Mamba-130M | 59% | 61% | 41% | 6% | 0% | 0% | 0% | 0% | 0% | 0% | 0% | 0% | 0% |
> | DeciMamba-130M | 70% | 70% | 53% | 48% | 51% | 58% | 67% | 68% | 73% | 60% | 61% | 53% | 21% |
>
> The setting is the same as the one in the paper. All models were trained on sequences of length 2k (10 documents). For each context length we test performance for 100 different queries.
>
> \
> **Zero-Shot Perplexity:**
> | Context Length | 1K | 2K | 4K | 8K | 10K | 20K | 30K | 40K | 50K | 60K | 70K | 80K |
> |---------|---------|-------------|------|------|-----|----|------|------|-----|----|------|------|
> | Pythia-2.8B | 10.24 | 9.96 | inf | inf | inf | inf | inf | inf |inf | inf | inf | inf |
> | Mamba-2.8B | 9.39 | 9.17 | 11.6 | inf | inf | inf | inf | inf |inf | inf | inf | inf |
> | DeciMamba-2.8b | 9.39 | 9.17 | 11.98 | 14.58 | 14.73 | 17.17 | 19.83 | 22.2 | 24.89 | 27.57 | 27.89 | 27.43 |
>
> | Context Length | 1K | 2K | 4K | 8K | 10K | 20K | 30K | 40K | 50K | 60K | 70K | 80K |
> |---------|---------|-------------|------|------|-----|----|------|------|-----|----|------|------|
> | Pythia-1.4B | 11.64 | 11.32 | inf | inf | inf | inf | inf | inf |inf | inf | inf | inf |
> | Mamba-1.4B | 10.51 | 10.31 | 10.5 | 14.43 | 18.4 | inf | inf | inf |inf | inf | inf | inf |
> | DeciMamba-1.4b | 10.51 | 10.31 | 10.5 | 14.13 | 14.5 | 18.33 | 23.54 | 26.82 | 28.97 | 30.56 | 29.97 | 29.28 |
>
> The setting is the same as the one in the paper. All models were pre-trained on sequences of length 2k (we remind that DeciMamba is applied directly without any tuning). Inf replaces any result larger than 100.
>
> For LongBench, the equivalent Transformer model (Pythia-2.8B) repeatedly causes an OOM error on our GPU (A6000, 48GB of RAM).
>
>
> \
> **Minor Point 5:**
> _________________________________________________________________________________________________________
> > Additional clarifications and edits
> _________________________________________________________________________________________________________
> We greatly appreciate the detailed feedback. We incorporated the suggested edits into the revised manuscript.
>
>
> \
> If all concerns are met, we kindly ask the reviewer to increment their score. Nevertheless, if any concerns remain unresolved, we would be happy to further address them.
>
> \
> **References:**  \
> [1] Long Range Arena: A Benchmark for Efficient Transformers, Tay et. al, ICLR ‘21 \
> [2] Never Train from Scratch: Fair Comparison of Long-Sequence Models Requires Data-Driven Priors, Amos et. al, ICLR ‘24 \
> [3] Simple linear attention language models balance the recall-throughput tradeoff, Arora et. al, ICLR ‘24 (WS)

---

### Author Response · Authors · 2024-11-24
**Global Response to Reviews and Manuscript Changes**

We sincerely thank all the reviewers for their detailed and constructive feedback, which helped improve our work. Based on the reviews, we have made several changes to further strengthen the paper:

**1. Hyperparameter Selection and Additional Ablation Studies:**
- Emphasized that DeciMamba's hyperparameter sweep is simple and short by clarifying the roles of each hyperparameter and providing additional ablation studies for the Decay Rate (beta) and Number of Decimation Layers hyperparameters.

**2. Additional Experimental Analysis:**
- Added a comparison with equivalent Transformer models across all tasks.
- By design, during inference, DeciMamba performs pooling only in the pre-fill stage. We present an experiment that demonstrates (i) the justification of this design, even in long generation tasks, and (ii) that extending our method to the decoding phase is straightforward.

**3. Method Explanation and Technical Details:**
- Improved the explanation of DeciMamba's method in the **Introduction** and in **Section 4**.
- Added a discussion of the factors affecting DeciMamba's success.
- Addressed the relationship between hidden state size and length generalization.
- Clarified that DeciMamba operates identically to Mamba in short-context tasks.
- Justified our selective pooling-based approach by citing a large body of relevant work from top-tier conferences and referencing additional relevant experiments in the appendix.
- Emphasized the depth and diversity of our experiments, which include multiple model sizes, datasets, and tasks, as well as efficiency benchmarking.

**4. Applications and Future Research:**
- Added future research directions aimed at improving additional sub-quadratic architectures.
- Added future research directions focused on gaining a better understanding of the limited ERF problem by utilizing our proposed **Mamba Mean Distance** metric.
- Cited a large body of work from top-tier conferences that further motivates the importance of improving the pre-fill phase.

**5. Extension to Other Architectures:**
- We introduced a new subsection titled **Extensions to Other Models** in Appendix E, which provides a detailed explanation of how the tools proposed in this paper can be adapted to other architectures.

\
The revised manuscript provides a more comprehensive and clearer presentation of our work while addressing all the concerns raised by the reviewers. New edits are highlighted in red.

---

### Meta-Review · Area_Chair_aMM9 · 2024-12-22

**Metareview:**

This paper studies the length generalization ability of Mamba and proposes a way to improve it. The paper proposes a measure named Mamba mean distance to capture the length of “effective receptive field (ERF)”, and identifies that Mamba extrapolating to longer sequence length suffers limited ERF, which leads to degraded performance. The authors then propose discarding tokens considered unimportant by Mamba, by choosing only tokens with large averaged discretization step size $\Delta_t$. Notably, the proposed method does not require re-training the model. The paper presents extensive numerical evaluation and shows consistent improvement of the proposed method DeciMamba over Mamba.

The paper tackles a fundamental problem in the study of SSMs and sequence processing models in general. Over diverse benchmarks and model size, the paper demonstrates consistent improvements in terms of context extension. Although the paper does not provide a complete solution to limited ERF, I view DeciMamba as a good “engineering solution” to improve existing Mamba models.

There were concerns raised and some reviewers were negative about the paper. One shortcoming of the proposed approach is that the proposed method only applies to the specific model Mamba, which is true. The authors clarified that while this is indeed the case, the ideas proposed in the paper can be applied to evaluate and improve other models (Appendix E). Another concern was that the proposed method only considers pre-filling stage, not decoding stage; the authors outlined ways to apply their method to the decoding stage and showed that applying DeciMamba for decoding yields limited benefit at least in the benchmark considered (Although I think the technique may become useful in tasks where we need to generate sequences longer than those seen in training).

Overall, the authors put extensive efforts during the rebuttal phase to respond to reviewers’ questions and reflect reviewers’ suggestions to the revised manuscript. The paper is now supplemented with richer contents, and I believe most of the issues raised by the reviewers were properly addressed. Hence, I recommend this paper for acceptance.

**Additional Comments On Reviewer Discussion:**

The paper received two reviews of score 3. Both reviewers did respond to authors’ initial rebuttals during the discussion period, but unfortunately did not respond to the follow-up. After a careful review of the discussion, my best judgment is that most concerns raised by the negative reviews were appropriately addressed by the authors in the responses and the revised submission.

---

### Decision · Program_Chairs · 2025-01-22

Accept (Poster)